# GLEAM: Learning to Match and Explain in Cross-View Geo-Localization

## Abstract

Cross-View Geo-Localization (CVGL) focuses on identifying correspondences between images captured from distinct perspectives of the same geographical location. However, existing CVGL approaches are typically restricted to a single view or modality, and their direct visual matching strategy lacks interpretability: they only determine whether two images correspond, without explaining the rationale behind the match. In this paper, we present GLEAM-C, a foundational CVGL model that unifies multiple views and modalities—including UAV imagery, street maps, panoramic views, and ground photographs—by aligning them exclusively with satellite imagery. Our framework enhances training efficiency through optimized implementation while achieving accuracy comparable to prior modality-specific CVGL models through a two-phase training strategy. Moreover, to address the lack of interpretability in traditional CVGL methods, we leverage the reasoning capabilities of multimodal large language models (MLLMs) to propose a new task, GLEAM-X, which combines cross-view correspondence prediction with explainable reasoning. To support this task, we construct a bilingual benchmark using GPT-4o and Doubao-1.5-Thinking-Vision-Pro to generate training and testing data. The test set is further refined through detailed human revision, enabling systematic evaluation of explainable cross-view reasoning and advancing transparency and scalability in geo-localization. Together, GLEAM-C and GLEAM-X form a comprehensive CVGL pipeline that integrates multimodal, multi-view alignment with interpretable correspondence analysis, unifying accurate cross-view matching with explainable reasoning and advancing **G**eo-**L**ocalization by enabling models to better **E**xplain **A**nd **M**atch. Code and datasets used in this work will be made publicly accessible.

## 1 Introduction

Cross-View Geo-Localization (CVGL) seeks to determine the geographic position of a query image by establishing correspondences with a geo-referenced database captured from alternative viewpoints, especially satellite observations(Chen et al., 2025b; Li et al., 2024b; Xia et al., 2024a). Unlike Global Navigation Satellite Systems (GNSS), which suffer from signal blockage and noise in urban canyons and complex environments, CVGL offers a complementary and robust vision-based localization paradigm. This capability has broad relevance to autonomous driving (Cui & Ge, 2003; Chen et al., 2023), robotic navigation (Nowak et al., 2024; Semborski & Idzkowski, 2024), unmanned aerial vehicles (UAVs) navigation (Xu et al., 2024; Suzuki et al., 2016; Wang et al., 2024), and augmented reality devices (Kamalam et al., 2022; Sathyanarayana et al., 2020), where reliable positioning forms the foundation for effective operation.

Despite its practical importance, CVGL remains a technically demanding problem. The task is complicated by the substantial appearance gap across heterogeneous viewpoints and modalities (Regmi & Shah, 2019; Ge et al., 2024). Consequently, prior studies have typically focused on a single viewpoint and modality, such as UAV imagery (Zheng et al., 2020), street maps (Zhao et al., 2021), panoramic views (Zhu et al., 2021), or ground photographs (Wu et al., 2024b). While these modality-specific approaches have achieved promising results, the problem of establishing robust cross-modal alignment within a unified framework remains unresolved. A unified CVGL framework offers several advantages. First, it enables multi-platform deployment through a single inference interface, reducing system complexity, memory footprint, and initialization latency on resource-constrained

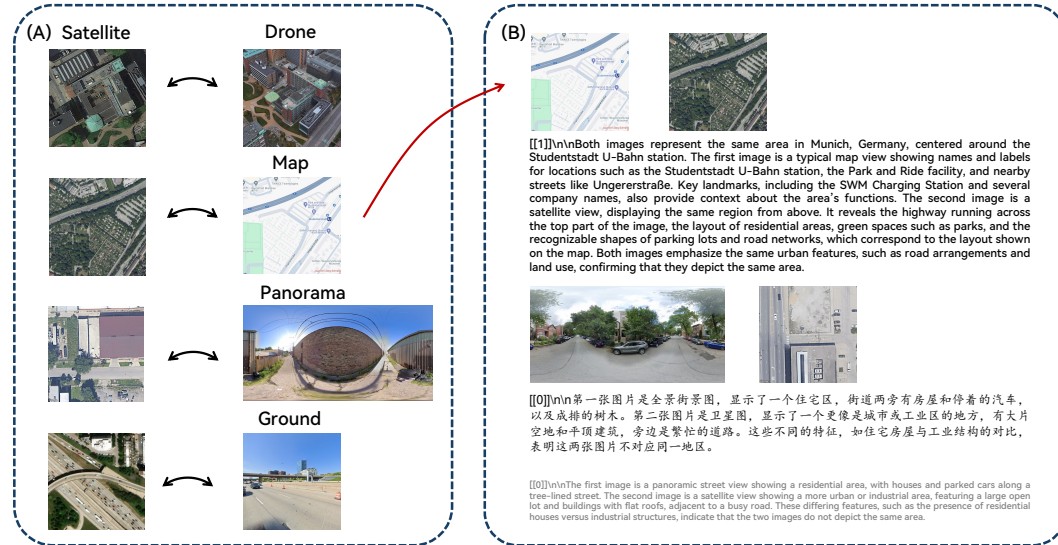

Figure 1: Overview of GLEAM-Core and GLEAM-eXplain. (A) GLEAM-C: a foundational CVGL model trained to align multiple views and modalities—including UAV imagery, street maps, panoramic images, and ground photos—with satellite imagery. (B) GLEAM-X: a benchmark combining cross-view correspondence prediction with explainable reasoning. We illustrate representative examples of both positive and negative matches between query images and satellite images. The red arrow indicates that GLEAM-C and GLEAM-X can be combined into an integrated pipeline.

platforms such as drones and vehicles. Second, it improves robustness and generalization by learning a geographically grounded representation that captures diverse structural regularities. By bridging heterogeneous views and modalities, such a system enables robust, generalizable, and efficient geo-positioning across diverse real-world scenarios.

Moreover, the dominant paradigm in CVGL has been image retrieval or binary correspondence prediction (Deuser et al., 2023a; Xia et al., 2024b), which offers limited interpretability and constrains system transparency. For safety-critical applications such as navigation or disaster response, a mere correspondence score is insufficient; instead, models must provide interpretable reasoning about why two views are matched or mismatched. Recent advances in multimodal large language models (MLLMs) present an avenue for addressing this limitation. By jointly processing visual and textual information, MLLMs can reason about image correspondences and generate human-interpretable explanations (Achiam et al., 2023; Bai et al., 2025a; Zhu et al., 2025). Integrating these reasoning capabilities into CVGL enables models not only to predict whether two images correspond but also to articulate the underlying rationale, enhancing transparency and accountability in decision-making.

To address these two challenges, in this paper, we propose a comprehensive solution that combines robust multi-modal alignment with interpretable cross-view reasoning. The overview of our work is shown in Fig. 1. First, we design GLEAM-Core (GLEAM-C), a foundational CVGL model that integrates multiple views and modalities—including UAV imagery, street maps, panoramic views, and ground photographs—by aligning them exclusively with satellite images. This unified design eliminates the need for modality-specific architectures, thereby simplifying the modeling process and improving scalability across heterogeneous data sources. In GLEAM-C, we reconstruct the conventional Data Parallel (DP) training strategy used in prior work (Deuser et al., 2023b) into a Distributed Data Parallel (DDP) scheme, which yields over 5-fold improvement in training efficiency. We design a two-phase training strategy and evaluate GLEAM-C on both CNN- and ViT-based architectures. The results show accuracy comparable to that of modality-specific models. Second, we design GLEAM-eXplain (GLEAM-X), which combines cross-view correspondence prediction with explainable reasoning. To enhance model robustness, we select 50k query images from the multi-view matching training set, each paired with a positive reference image (match) and a negative reference image (mismatch). After assigning ground-truth labels, GPT-4o and Doubao-1.5-Thinking-Vision-Pro generate explanations in both Chinese and English, clarifying the reasons for image correspondences. For evaluation, we select 504 query images from the matching test set, each paired with a positive and a negative reference image. After GPT-4o generates explanations, we perform multiple rounds of human annotation and refinement to create a high-quality test set. We fine-tune the Qwen2.5-VL-3B-Instruct MLLM on the training set, achieving higher matching accuracy than GPT-4o and Doubao-1.5, with analysis results closely aligned with human annotations. GLEAM-C and GLEAM-X can be further integrated: GLEAM-C performs the core task of cross-

view geo-localization by aligning query images with geo-referenced satellite imagery. GLEAM-X enhances this process by verifying and providing human-interpretable explanations for the image correspondences predicted by GLEAM-C, improving both the model's robustness and transparency. The contributions of our work can be summarized as follows:

**1) Problem Analysis**: We provide a detailed analysis of the CVGL task, highlight the benefits of a unified CVGL framework, and identify the lack of interpretability in current CVGL solutions.

**2) GLEAM-C**: We propose GLEAM-C, a foundational CVGL model that unifies multiple views and modalities—including UAV images, street maps, panoramic images, and ground photographs—by aligning them with satellite images. We adopt a two-phase training strategy and leverage DDP to optimize training efficiency, achieving accuracy comparable to modality-specific CVGL models.

**3) GLEAM-X**: We introduce the GLEAM-X benchmark, combining cross-view correspondence prediction with explainable reasoning. We provide a high-quality bilingual (Chinese-English) training and testing dataset, along with the Qwen2.5-VL-3B-Instruct model fine-tuned on this dataset. This enables the generation of human-interpretable explanations for image correspondences, thereby enhancing the robustness and transparency of the model.

**4) Integrated Pipeline**: GLEAM-C and GLEAM-X can be combined into an integrated pipeline. GLEAM-C performs the core task of CVGL by aligning query images with satellite imagery, while GLEAM-X enhances this process by verifying and providing human-interpretable explanations for the predicted correspondences, improving both the model's robustness and transparency.

## 2 RELATED WORKS

### 2.1 CROSS-VIEW GEO-LOCALIZATION

Cross-View Geo-Localization (CVGL) has garnered substantial attention within the research community. Most existing research, however, focuses on a single view or modality. Early efforts construct paired ground-to-aerial datasets (Lin et al., 2015), which later evolve into widely used benchmarks such as CVUSA (Workman et al., 2015), CVACT (Liu & Li, 2019), VIGOR (Zhu et al., 2021), University-1652 (Zheng et al., 2020), and DenseUAV (Dai et al., 2023), enabling evaluations across panoramas, UAV imagery, and related scenarios. Methodologically, handcrafted descriptors (Bansal et al., 2011; Castaldo et al., 2015) soon give way to deep learning, where pre-trained CNNs and fine-tuning strategies significantly advance cross-view correspondence (Krizhevsky et al., 2012; Workman et al., 2015). Subsequent studies introduce a variety of techniques to alleviate viewpoint discrepancies, including polar and optimal transport transformations (Shi et al., 2020; 2019), region-level and latent alignment (Dai et al., 2021; Xia et al., 2024a), and strategies that enhance scene discrimination, such as hard-negative mining (Deuser et al., 2023b) and cross-dimension interactions (Shen et al., 2023). More recently, unsupervised approaches explore how to exploit unlabeled data for training (Li et al., 2024c;b). Despite these advancements, most studies remain modality-specific, and the development of a unified framework that accommodates diverse modalities still represents an open challenge.

### 2.2 MULTIMODAL LARGE LANGUAGE MODEL

Multimodal large language models (MLLMs) have emerged in recent years as a powerful tool for handling complex tasks by incorporating visual inputs into traditional language models (OpenAI, 2023; Liu et al., 2024; Chen et al., 2024; Bai et al., 2025a). Prominent commercial models such as OpenAI's GPT-4o (Hurst et al., 2024) and Google's Gemini 2.5 (Comanici et al., 2025) exemplify this trend by simultaneously handling text, audio, images, and video, unlocking new capabilities like real-time, human-like voice interaction and long-context video understanding. In addition to closed-source commercial MLLMs, numerous well-known open-source models, such as InternVL3 (Wang et al., 2025), Qwen2.5-VL (Bai et al., 2025b), MiniCPM-V 4.5 (Yao et al., 2024), and LLaVA-OneVision (Li et al., 2024a), have made significant contributions to advancing multimodal understanding. Beyond perception alone, recent efforts increasingly emphasize unifying multimodal understanding and generation, thereby enabling models not only to interpret visual inputs but also to synthesize new content. Representative examples include Janus-Pro (Chen et al., 2025a), Show-

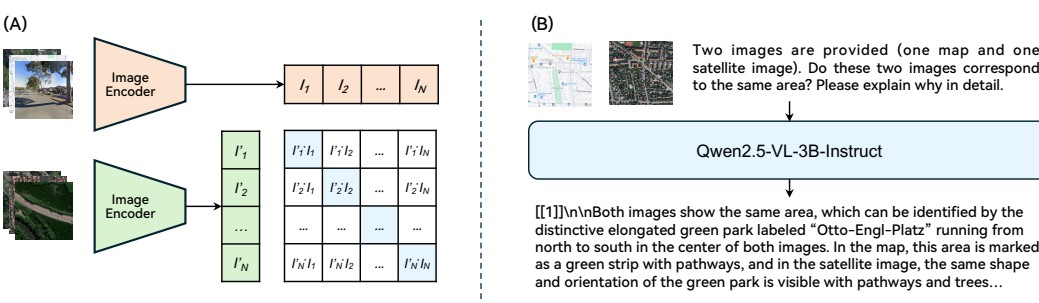

Figure 2: Method Overview of GLEAM-C and GLEAM-X. (A) GLEAM-C: We apply a contrastive learning architecture to train the CVGL model across UAV, street map, panoramic, and ground photographs. (B) GLEAM-X: This component formulates a multi-image reasoning task. The MLLM receives a query image, a reference image, and a natural language instruction. Through fine-tuning, it delivers both a matching prediction and an interpretable textual explanation.

| Dataset | Original Pairs | GLEAM-C | | GLEAM-X | |
|---|---|---|---|---|---|
| | | Sampling Ratio | Training Pairs | Sampling Ratio | Training Pairs (pos+neg, en+zh) |
| University-1652 (Zheng et al., 2020) | 37,854 | 1.00 | 37,854 | 0.36 | 54,000 |
| VIGOR (Zhu et al., 2021) | 52,609 | 1.00 | 52,609 | 0.26 | 54,000 |
| SetVL-480K (Wu et al., 2024a) | 240,544 | 0.50 | 120,272 | 0.06 | 54,000 |
| MAP | 10,208 | 4.00 | 40,832 | 1.00 | 40,832 |
| **Total** | **341,215** | — | **251,567** | — | **202,832** |

Table 1: Statistics of training pairs in GLEAM-C and GLEAM-X. GLEAM-C primarily constructs training pairs from existing CVGL datasets. GLEAM-X balances the data distribution across datasets and then samples positive and negative pairs, providing bilingual explanation annotations.

o (Xie et al., 2024), and BAGEL (Deng et al., 2025), which collectively mark a transition toward more general-purpose multimodal intelligence that integrates both reasoning and creativity.

## 3 GLEAM-C: A UNIFIED CVGL CORE MODEL

GLEAM-C is designed as a unified CVGL model, leveraging a contrastive learning paradigm to align visual representations across diverse viewpoints and modalities. In this section, we provide the data composition (Sec. 3.1), model structure (Sec. 3.2), and training recipe (Sec. 3.3).

### 3.1 DATA COMPOSITION

GLEAM-C is a foundational CVGL model that aligns multiple views and modalities—including UAV imagery, street maps, panoramic views, and ground photographs—exclusively with satellite imagery. To facilitate training, we utilize widely adopted datasets from prior work: University-1652 (Zheng et al., 2020) for UAV imagery, VIGOR (Zhu et al., 2021) for panoramic views, and SetVL-480K (Wu et al., 2024a) for ground photographs. In addition, for street maps, we manually collect 12,761 pairs of corresponding street maps and satellite images from Google Maps, with each image resized to 512×512 pixels. We use 10,208 pairs for training and 2,553 pairs for testing.

The original datasets show highly imbalanced sample distributions across different views and modalities, as shown in Tab. 1, which could introduce biases and hinder the model from learning uniformly. To mitigate this issue, we employ data sampling to balance the number of samples across all views and modalities. This ensures that the model receives sufficient and diverse training signals from each type of data, thereby improving its robustness across heterogeneous inputs. For evaluation, we adhere to the original test datasets and protocols provided by each benchmark. The newly introduced MAP test follows the same evaluation logic as VIGOR (Zhu et al., 2021).

### 3.2 MODEL STRUCTURE

We leverage a contrastive training strategy for GLEAM-C, where the query and reference images share the same encoder for feature extraction. The InfoNCE loss is applied to learn discriminative features across both view directions. The training framework is shown in Fig. 2A. Our implementation is based on the well-engineered Sample4Geo (Deuser et al., 2023b) codebase. Notably, Sample4Geo introduces GPS-Sampling (GPS) and Dynamic Similarity Sampling (DSS) to select hard negatives during training, thereby improving optimization efficiency. However, certain datasets,

such as University-1652 (Zheng et al., 2020), do not provide GPS metadata. Therefore, GLEAM-C relies solely on DSS. In our experiments, we evaluate both convolutional and transformer-based architectures as backbones. Specifically, we adopt ConvNeXt (Liu et al., 2022), comprising 88.6M parameters, and the Perception Encoder (Bolya et al., 2025), comprising 0.32B parameters.

### 3.3 TRAINING RECIPE

We propose a two-phase training scheme to effectively integrate knowledge across multiple modalities and views (Sec. 3.3.1). Given the substantial data requirements of foundational model training and the limited efficiency of the original Sample4Geo codebase, we implement targeted optimizations to significantly enhance training efficiency (Sec. 3.3.2).

#### 3.3.1 TWO-PHASE TRAINING STRATEGY

We note that a naive concatenation of multiple datasets for training from the outset poses significant challenges. Due to the backbone model's limited matching capacity and the varying dataset sizes and difficulty levels, direct multi-dataset training often results in highly imbalanced performance. To mitigate this issue, we first train the model on a single dataset that is relatively large and of moderate difficulty, allowing the backbone to acquire fundamental CVGL capabilities before introducing more complex and diverse data. In our design, we first train the model on the VIGOR (Zhu et al., 2021) dataset for 40 epochs, followed by training on the concatenated datasets for an additional 40 epochs.

#### 3.3.2 TRAINING EFFICIENCY IMPROVEMENT

Similar to CLIP training, contrastive learning models typically require large batch sizes. The original Sample4Geo adopts PyTorch's Data Parallel (DP) strategy, where GPU 0 distributes data, aggregates outputs, and performs gradient updates for the entire cluster. This centralized design creates a significant communication bottleneck on GPU 0, thus slowing the training process. Distributed Data Parallel (DDP) alleviates this issue by allowing each GPU to compute gradients locally and synchronizing them via an efficient all-reduce. However, contrastive learning relies on in-batch negatives, and small per-GPU batch sizes limit the number of negatives available per device, potentially degrading performance. This underscores the importance of maintaining sufficiently large per-GPU batch sizes even when using DDP. To tackle this issue, we adopt the strategy implemented in Open-CLIP (Ilharco et al., 2021). Before computing the contrastive loss, each GPU (rank) gathers the feature representations from all other GPUs. These features are then re-concatenated, effectively creating a unified global batch for the loss calculation on each device. This process preserves the gradient flow for proper backpropagation while providing every GPU with a much larger and more effective set of negative samples. Due to page limitations, we provide pseudocode for the loss computation in Alg. 1 in Sec. A.1. This implementation achieves over 5-fold faster training speed while maintaining the original model accuracy.

## 4 GLEAM-X: A MULTI-IMAGE MLLM EXPLANATION BENCHMARK

GLEAM-X extends GLEAM-C to a multi-image reasoning setting, where the model not only predicts matches between query and reference images but also generates interpretable explanations. In this section, we present the data composition (Sec. 4.1), annotation procedure (Sec. 4.2), evaluation protocol (Sec. 4.3), and how it can be combined with GLEAM-C as an integrated pipeline (Sec. 4.4).

### 4.1 DATA COMPOSITION

Our training and testing data pairs are derived from the GLEAM-C training and test sets. To maintain balance across modalities and views, we perform data sampling to ensure that the amount of training data for each modality is roughly equal, as shown in Tab. 1. In the evaluation of GLEAM-X, we also maintain this balance across modalities. The selected query images for testing include 128 for MAP, 126 for SetVL-480K, 127 for University-1652, and 123 for VIGOR. For each query image, we select one positive reference image and one negative reference image. Using the corresponding labels, we generate explanation annotations in both English and Chinese with commercial MLLMs, specifically GPT-4o (Hurst et al., 2024) and Doubao-1.5-Thinking-Vision-Pro (Volcengine, 2025). Notably, to ensure linguistic consistency and semantic equivalence in the bilingual benchmark, we

generate both language versions simultaneously in a single prompt, explicitly requiring the model to provide corresponding explanations in English and Chinese.

During explanation construction, the commercial MLLMs are provided with ground-truth labels indicating whether a given query-reference pair matches, guiding them to generate explanations accordingly. This introduces explicit prior knowledge, thereby enhancing the correctness of the generated responses. During MLLM training, the ground-truth label is concatenated with the explanations produced by the commercial MLLMs and used as the target output. For example, for a positive sample pair, the model is trained to produce the response `[[1]]\n\n(explanation)`. Such a format enables straightforward extraction of match labels, supporting both training supervision and evaluation of matching performance. Using this approach, we construct 200k training pairs with GPT-4o and Doubao-1.5-Thinking-Vision-Pro, respectively, and 2k test pairs with GPT-4o. The architecture employed for MLLM training and inference is shown in Fig. 2B.

### 4.2 Data Annotation

To ensure the accuracy of the test set, we manually revise the 2k test pairs. We engage 4 human experts in Remote Sensing, each holding at least a master's degree, to perform data annotation and revision. We conduct two annotation rounds: in each round, we first verify the correctness of the explanations generated by the commercial MLLMs and correct them if necessary; if the four experts cannot determine whether a query and reference image match, we replace the pair with a suitable one from the remaining test data and perform model-assisted generation followed by manual correction. This ensures that all test pairs are valid and unambiguous, providing a reliable basis for evaluation.

### 4.3 Evaluation Protocol

We evaluate the MLLM outputs on the test set from two perspectives. First, we extract the predicted match labels using regular-expression parsing and compute the matching accuracy. Second, we assess the quality of the generated explanations by measuring their semantic similarity to the annotated references with Sentence-BERT (Reimers & Gurevych, 2019). This dual evaluation framework ensures a comprehensive assessment, as it captures both the correctness of structured match predictions and the semantic fidelity of explanatory outputs.

### 4.4 GLEAM-C and GLEAM-X as an Integrated Pipeline

In real-world deployment, GLEAM-C and GLEAM-X operate as a two-stage pipeline that combines efficient image retrieval with interpretable verification, as illustrated in Fig. 1. Given a query image and a reference database, GLEAM-C first retrieves the most similar reference image through cross-view geo-localization. GLEAM-X then verifies whether the query and retrieved reference correspond to the same geographic location, while providing human-interpretable explanations that identify key visual correspondences or discrepancies between the image pair. This integrated approach not only enhances retrieval reliability but also offers transparency through explainable reasoning, which is critical for safety-critical real-world applications such as autonomous navigation, search and rescue operations, and emergency response.

## 5 Experiments

In this section, we design a series of experiments to evaluate the unified GLEAM-C model (Sec. 5.1) and the GLEAM-X benchmark (Sec. 5.2). All training experiments are conducted on a cluster equipped with 10 NVIDIA RTX 4090 D GPUs, each with 48 GB of memory. In addition, to assess practical deployment feasibility, we evaluate the models' inference performance on an NVIDIA Jetson AGX Xavier for real-world edge computing scenarios in the Appendix (Sec. A.4).

### 5.1 Evaluation results of GLEAM-C

For GLEAM-C, we compare the unified model with single-dataset training and evaluate the data integration strategy (Sec.5.1.1). We further assess improvements in training efficiency (Sec.5.1.2) relative to prior SOTA training methods. Additionally, we investigate cross-domain generalization performance and the rationale for selecting VIGOR as the initial training dataset (Sec. 5.1.3).

| Backbone | Training Strategy | Dataset | Epochs | R@1 | R@5 | R@10 | Top-1 | AP/HR |
|---|---|---|---|---|---|---|---|---|
| **ConvNeXt-B-384** | Single-dataset **Sample4Geo** †**GPS + DSS** | †VIGOR | 40 | 77.86 | 95.66 | 97.21 | 99.61 | 89.82 |
| | | University-1652 | 1 | 92.65 | - | - | - | 93.81 |
| | | †SetVL-480K | 40 | 16.86 | 38.95 | 47.71 | 85.40 | 16.86 |
| **ConvNeXt-B-384** | Single-dataset **Sample4Geo w/o GPS** | VIGOR | 40 | 76.60 | 95.07 | 96.83 | 99.64 | 88.29 |
| | | University-1652 | 1 | 91.55 | 97.79 | 98.37 | 98.44 | 92.98 |
| | | MAP | 40 | 92.60 | 98.08 | 98.94 | 99.80 | 92.60 |
| | | SetVL-480K | 40 | 14.23 | 34.10 | 44.63 | 71.67 | 14.23 |
| | From-scratch (Merge) | VIGOR | 40 | 73.37 | 93.43 | 95.66 | 99.61 | 84.43 |
| | | University-1652 | | 87.08 | 95.22 | 96.59 | 96.79 | 88.96 |
| | | MAP | | 92.52 | 97.81 | 98.79 | 99.80 | 92.52 |
| | | SetVL-480K | | 14.33 | 34.30 | 44.77 | 70.75 | 14.33 |
| | Two-phase (VIGOR → Merge) | VIGOR | 40 | 75.66 | 94.51 | 96.43 | 99.65 | 86.80 |
| | | University-1652 | | 87.03 | 96.11 | 97.84 | 98.03 | 89.12 |
| | | MAP | | 94.05 | 98.04 | 98.71 | 99.65 | 94.05 |
| | | SetVL-480K | | 15.28 | 35.49 | 45.71 | 71.30 | 15.28 |
| **PE-Core-L14-336** | Single-dataset | VIGOR | 40 | 75.53 | 95.48 | 97.18 | 99.67 | 89.46 |
| | | University-1652 | 1 | 94.38 | 98.44 | 98.80 | 98.85 | 95.32 |
| | | MAP | 40 | 92.79 | 97.92 | 98.55 | 99.73 | 92.79 |
| | | SetVL-480K | 40 | 21.34 | 46.54 | 57.95 | 81.58 | 21.34 |
| | From-scratch (Merge) | VIGOR | 40 | 69.44 | 92.81 | 95.43 | 99.63 | 83.18 |
| | | University-1652 | | 93.20 | 97.73 | 98.30 | 98.40 | 94.26 |
| | | MAP | | 93.11 | 98.32 | 98.75 | 99.73 | 93.11 |
| | | SetVL-480K | | 22.00 | 46.93 | 58.28 | 81.83 | 22.00 |
| | Two-phase (VIGOR → Merge) | VIGOR | 40 | 75.96 | 95.46 | 97.17 | 99.68 | 89.44 |
| | | University-1652 | | 93.19 | 97.92 | 98.49 | 98.57 | 94.28 |
| | | MAP | | 93.97 | 98.20 | 98.43 | 99.77 | 93.97 |
| | | SetVL-480K | | 23.25 | 48.61 | 59.76 | 82.98 | 23.25 |

Table 2: Image retrieval accuracy across backbones and training strategies. For University-1652, the last evaluation metric is AP; for all other datasets, it is Hit Rate. The first three rows (highlighted in gray) are directly taken from the corresponding papers with GPS (if applicable, marked with †) and DSS sampling for reference. It can be observed that using a two-phase training strategy yields relatively stronger performance. The unified GLEAM-C model achieves results comparable to or even exceeding those obtained by single-dataset training (Sample4Geo w/o GPS-Sampling).

| Dataset | Mode | Recall@1 | Recall@5 | Recall@10 | Recall@top1 | Hit Rate | Seconds/Epoch |
|---|---|---|---|---|---|---|---|
| **VIGOR** | DP | 76.80 | 95.06 | 96.70 | 99.25 | 89.05 | 521 |
| | DDP | 76.60 | 95.07 | 96.83 | 99.64 | 88.29 | 92 |

Table 3: Image retrieval accuracy comparison between Data Parallel (DP) and Distributed Data Parallel (DDP) on the VIGOR dataset with ConvNeXt-B backbone using Sample4Geo codebase. Both approaches achieve comparable accuracy, while DDP attains over 5× faster training speed.

### 5.1.1 FROM SINGLE DATASET TO MULTI-DATASET INTEGRATION

As in most CVGL studies, we first train on single-view and single-modality datasets. Training is conducted separately on the VIGOR (same), University-1652, MAP, and SetVL-480K (N = 1) datasets. Model performance is evaluated using image retrieval accuracy, with University-1652 following the Drone2Sat protocol. We adopt ConvNeXt-B-384 (88.6M) and PE-Core-L14-336 (0.32B) as the image encoders. The results for single-dataset training are presented in Tab. 2. Since some datasets lack GPS metadata, we do not use GPS sampling in all our model training, which may result in a slight decrease in accuracy. For reference, we also report the original metrics from the respective papers, with GPS (if applicable) and DSS sampling (Deuser et al., 2023b; Wu et al., 2024a).

To train the unified model, we compare two data integration strategies in our work. The first strategy trains on mixed data directly from the backbone's initial weights, while the second strategy starts training from a VIGOR checkpoint obtained through single-dataset training. We use the data sampling ratios shown in Tab. 1, and both strategies are trained for 40 epochs. We provide the results in Tab. 2 and share the following observations:

**1)** Directly training on the mixed data degrades the performance on VIGOR. However, continuing training the unified model from a VIGOR checkpoint yields better results and improves accuracy on MAP and SetVL-480K. This highlights the importance of endowing the model with basic CVGL capability before performing mixed-data training, and also suggests that learning across different views and modalities may mutually reinforce each other.

**2)** Compared with single-dataset training, our two-phase training strategy yields the foundational GLEAM-C model, which attains comparable or superior performance across all evaluated datasets.

**3)** The larger PE-Core-L14-336 model achieves overall better performance compared to ConvNeXt-B-384 after training. Additionally, although University-1652 requires only a single epoch of training, we train for 40 epochs due to the demands of mixed-data training; under this setting, smaller models may suffer from overtraining or instability, whereas larger models remain largely unaffected.

### 5.1.2 TRAINING EFFICIENCY EVALUATION

We evaluate training efficiency on the VIGOR dataset by comparing DP and DDP strategies. Following the experiment settings of Sample4Geo (Deuser et al., 2023b), we adopt the ConvNeXt-B-384 backbone and use the original Sample4Geo code for DP training (without GPS-based sampling). As shown in Tab. 3, both approaches achieve comparable image retrieval accuracy across all Recall@k and Hit Rate metrics. However, DDP demonstrates a substantial (5-fold) speed advantage, reducing the time per training epoch from 521 seconds to 92 seconds, highlighting its effectiveness for accelerating large-scale CVGL model training.

### 5.1.3 DISCUSSION: CROSS-DOMAIN GENERALIZATION & INITIAL DATASET SELECTION

Beyond the above analysis, we investigate two important questions regarding our unified model training scheme: **1)** What is the cross-domain generalization performance when trained on one modality and tested on unseen modalities? **2)** Why do we choose VIGOR as the initial dataset in our two-phase training strategy? Due to space constraints, detailed experimental analysis and results are provided in Sec. A.5 and Sec. A.6, respectively.

## 5.2 EVALUATION RESULTS ON GLEAM-X

We evaluate both commercial models and open-source models on the GLEAM-X benchmark. For commercial models, we regenerate answers on the test set using GPT-4o and Doubao-1.5-Thinking-Vision-Pro without providing ground-truth labels. For open-source models, we compare three variants of Qwen2.5-VL-3B-Instruct: the original model, the version fine-tuned with LLM-explanation supervision, and a version trained solely with label-only supervision. Specifically, for the explanation-based variant, we follow the procedure in Sec. 4.1, concatenating labels and explanations during training. We train separately on the 200k explanations generated by GPT-4o and Doubao-1.5 for 1 epoch. Conversely, the label-only variant uses only labels (e.g., [[1]]) for training.

After inference on the test set, we decouple matching predictions and explanations through regularization to isolate their respective contributions. To provide a comprehensive evaluation of GLEAM-X, we perform detailed analyses from three perspectives: matching accuracy (Sec.5.2.1), semantic accuracy (Sec.5.2.2), and positive/negative sample analysis (Sec. 5.2.3). Additionally, we explore alternative semantic evaluation methods beyond Sentence-BERT (Sec. 5.2.4) and discuss potential biases in dataset curation (Sec. 5.2.5) to ensure a more thorough analysis.

### 5.2.1 MATCHING ACCURACY

The matching accuracy results are presented in Tab. 4. Commercial models achieve around 80% accuracy, with GPT-4o outperforming Doubao-1.5 on average. In contrast, the original Qwen2.5-VL-3B-Instruct model exhibits very low accuracy, particularly in Chinese (50.50%). After training, all model variants surpass commercial MLLMs. Notably, the label-only supervision model achieves the highest accuracy, likely because its training targets are extremely short (e.g., [[0]] or [[1]]), which reduces learning difficulty. By comparison, the explanation-supervised model must learn the harder task of generating full reasoning while making predictions, resulting in relatively lower accuracy. Despite this, it still substantially outperforms the baseline and can explain its decisions. This reflects a design choice where GLEAM-X prioritizes interpretability over maximizing accuracy alone.

### 5.2.2 SEMANTIC ANALYSIS OF EXPLANATIONS

Tab. 5 reveals the similarity distributional differences across models. Commercial MLLMs concentrate heavily in the high-similarity range (0.8–1.0) for English, but shift toward 0.6–0.8 in Chinese.

| Model | Language | MAP | SetVL-480K | University-1652 | VIGOR | Avg Acc |
|---|---|---|---|---|---|---|
| GPT-4o | | 85.55 | 63.89 | 93.70 | 81.71 | 81.25 |
| Doubao-1.5 | | 91.80 | 74.21 | 60.24 | 83.74 | 77.48 |
| Qwen2.5-VL-3B-Instruct | EN | 55.08 | 57.94 | 51.57 | 65.04 | 57.34 |
| +label only | | 96.88 | 84.13 | 97.64 | 92.28 | 92.76 |
| +GPT-4o | | 94.53 | 78.17 | 99.61 | 89.02 | 90.38 |
| +Doubao-1.5 | | 95.31 | 79.37 | 96.46 | 87.80 | 89.78 |
| **Model** | **Language** | **MAP** | **SetVL-480K** | **University-1652** | **VIGOR** | **Avg Acc** |
| GPT-4o | | 85.55 | 70.63 | 96.06 | 82.11 | 83.63 |
| Doubao-1.5 | | 89.45 | 78.97 | 66.93 | 81.30 | 79.17 |
| Qwen2.5-VL-3B-Instruct | ZH | 51.17 | 51.19 | 50.00 | 49.59 | 50.50 |
| +label only | | 96.88 | 84.13 | 97.64 | 92.28 | 92.76 |
| +GPT-4o | | 95.31 | 76.59 | 99.61 | 89.02 | 90.18 |
| +Doubao-1.5 | | 94.53 | 75.00 | 94.88 | 86.59 | 87.80 |

Table 4: Matching accuracy comparison of different models on GLEAM-X in English and Chinese. Results are reported on MAP, SetVL-480K, University-1652, and VIGOR datasets, along with the overall average accuracy. Label-only supervision and LLM-explanation supervision (+GPT-4o, +Doubao-1.5) significantly improve the performance of Qwen2.5-VL-3B-Instruct.

| Model | Language | 0.0-0.2 | 0.2-0.4 | 0.4-0.6 | 0.6-0.8 | 0.8-1.0 | Avg Sim |
|---|---|---|---|---|---|---|---|
| GPT-4o | | 0.0% | 0.1% | 0.8% | 21.6% | 77.5% | 0.8349 |
| Doubao-1.5 | | 0.0% | 0.0% | 0.2% | 33.5% | 66.3% | 0.8171 |
| Qwen2.5-VL-3B-Instruct | EN | 0.8% | 0.0% | 2.0% | 59.3% | 37.9% | 0.7645 |
| +label only | | 99.9% | 0.1% | 0.0% | 0.0% | 0.0% | 0.0981 |
| +GPT-4o | | 0.0% | 0.0% | 0.1% | 22.4% | 77.5% | 0.8405 |
| +Doubao-1.5 | | 0.0% | 0.0% | 0.5% | 36.0% | 63.5% | 0.8122 |
| **Model** | **Language** | **0.0-0.2** | **0.2-0.4** | **0.4-0.6** | **0.6-0.8** | **0.8-1.0** | **Avg Sim** |
| GPT-4o | | 1.6% | 5.0% | 2.3% | 50.5% | 40.7% | 0.7403 |
| Doubao-1.5 | | 0.0% | 0.0% | 4.5% | 66.7% | 28.9% | 0.7495 |
| Qwen2.5-VL-3B-Instruct | ZH | 4.2% | 0.1% | 2.9% | 58.5% | 34.3% | 0.7399 |
| +label only | | 92.1% | 7.9% | 0.0% | 0.0% | 0.0% | 0.1476 |
| +GPT-4o | | 0.0% | 0.0% | 0.3% | 32.5% | 67.2% | 0.8239 |
| +Doubao-1.5 | | 0.4% | 0.0% | 1.1% | 46.6% | 51.9% | 0.7924 |

Table 5: Similarity score distribution (by Sentence-BERT) for different models on GLEAM-X in English and Chinese. Columns show percentages in each similarity range (0.0–0.2, ..., 0.8–1.0) and the last column reports Avg Sim. LLM-explanation supervision (+GPT-4o, +Doubao-1.5) improves alignment with ground-truth, while label-only supervision fails to generate explanations.

The original Qwen2.5-VL-3B-Instruct places a substantial proportion of predictions in the mid-similarity bins (0.6–0.8) for both languages, indicating weaker semantic alignment with ground-truth explanations. Fine-tuning with LLM-explanation supervision shifts the distribution upward, with both +GPT-4o and +Doubao-1.5 variants peaking in the 0.8–1.0 range. By contrast, the label-only supervision model collapses almost entirely into the lowest bin (0.0–0.2), confirming its inability to generate semantically meaningful explanations despite improvements in matching accuracy.

### 5.2.3 ANALYSIS OF POSITIVE AND NEGATIVE PAIR SAMPLES

To further analyze model behavior on positive and negative samples, we examine both matching accuracy and semantic similarity in Tab. 6. Vanilla MLLMs, including both commercial models and Qwen2.5-VL-3B-Instruct, exhibit a bias toward negative samples. In particular, Qwen2.5-VL-3B-Instruct achieves very low positive accuracy while maintaining high negative accuracy, indicating that the original model tends to predict image pairs as non-matching. The negative bias stems from the inherent asymmetry in the task: identifying mismatches requires finding just one distinguishing feature, while confirming matches demands comprehensive evidence across multiple aspects.

Fine-tuning with LLM-explanation supervision (+GPT-4o, +Doubao-1.5) substantially mitigates this bias, raising positive accuracy to over 90% in English and over 84% in Chinese, while still keeping negative accuracy high. Correspondingly, positive and negative similarity are more balanced. Overall, LLM-explanation supervision effectively improves both prediction balance and explanation quality across languages. In contrast, the label-only supervision variant achieves high positive accuracy but fails to generate semantically meaningful explanations, as evidenced by extremely low similarity scores.

| Model | Language | Pos Acc | Neg Acc | Pos-Neg | Pos Sim | Neg Sim | Pos-Neg |
|---|---|---|---|---|---|---|---|
| GPT-4o | | 70.83 | 91.67 | -20.84 | 0.8296 | 0.8401 | -0.0105 |
| Doubao-1.5 | | 66.87 | 88.10 | -21.23 | 0.8100 | 0.8241 | -0.0141 |
| Qwen2.5-VL-3B-Instruct | EN | 23.81 | 90.87 | -67.06 | 0.7616 | 0.7675 | -0.0059 |
| +label only | | 93.85 | 91.67 | 2.18 | 0.0851 | 0.1111 | -0.0260 |
| +GPT-4o | | 93.25 | 87.50 | 5.75 | 0.8399 | 0.8411 | -0.0012 |
| +Doubao-1.5 | | 90.08 | 89.48 | 0.60 | 0.8048 | 0.8195 | -0.0147 |
| **Model** | **Language** | **Pos Acc** | **Neg Acc** | **Pos-Neg** | **Pos Sim** | **Neg Sim** | **Pos-Neg** |
| GPT-4o | | 73.02 | 94.25 | -21.23 | 0.7337 | 0.7468 | -0.0131 |
| Doubao-1.5 | | 76.19 | 82.14 | -5.95 | 0.7435 | 0.7555 | -0.0120 |
| Qwen2.5-VL-3B-Instruct | ZH | 1.19 | 99.80 | -98.61 | 0.7265 | 0.7533 | -0.0268 |
| +label only | | 93.85 | 91.67 | 2.18 | 0.1471 | 0.1481 | -0.0010 |
| +GPT-4o | | 90.87 | 89.48 | 1.39 | 0.8211 | 0.8267 | -0.0056 |
| +Doubao-1.5 | | 84.52 | 91.07 | -6.55 | 0.7858 | 0.7989 | -0.0131 |

Table 6: Positive and negative sample accuracy (Pos/Neg Acc) and similarity (Pos/Neg Sim) for different models on GLEAM-X in English and Chinese. The Pos-Neg columns report the difference between positive and negative samples. Vanilla Qwen2.5-VL-3B-Instruct shows a strong bias toward predicting non-matching pairs, while LLM-explanation supervision (+GPT-4o, +Doubao-1.5) mitigates this bias and improves alignment with ground-truth explanations.

### 5.2.4 MORE SEMANTIC EVALUATION METHODS

For semantic analysis, while Sentence-BERT offers low validation costs in terms of both price and computational efficiency, it is limited to measuring textual similarity between model outputs and annotations, potentially failing to capture critical reasoning steps grounded in the visual content itself. To address this limitation, we supplement our evaluation with an LLM-as-a-judge approach using the multimodal Gemini 2.5 Flash (Comanici et al., 2025) model, which can assess both textual reasoning and visual matching. Furthermore, we conduct human evaluation to validate the effectiveness of both automated assessment methods. Our experiments demonstrate strong alignment across all three evaluation approaches (detailed in Sec. A.7).

### 5.2.5 DISCUSSION: POTENTIAL BIASES IN DATASET CURATION

GLEAM-X relies on GPT-4o and Doubao-1.5 to generate explanation labels, which may introduce biases such as linguistic patterns or superficial reasoning into the training data, raising concerns about propagating teacher model limitations to student models.

Our approach mitigates these concerns through ground-truth guidance during data curation. When generating explanations, we provide GT matching labels to teacher models, steering them toward correct reasoning directions. As shown in Tab. 4 and Tab. 5, teacher models (GPT-4o and Doubao-1.5) achieve 77-84% average accuracy and 0.74-0.83 average similarity when evaluated without GT labels during inference. In contrast, student models (Qwen2.5-VL-3B) trained on GT-guided explanations achieve 87-91% accuracy and 0.79-0.84 similarity, demonstrating that GT supervision effectively controls bias and enables students to even surpass teacher performance.

The quality of generated explanations depends on the teacher model's inherent capabilities, which determine the ceiling for knowledge transfer. In this work, we have not performed additional data refinement (e.g., human filtering or iterative improvement). Nevertheless, the strong performance indicates that the current explanation quality is sufficient for effective knowledge transfer. As this is the first work to introduce explanation supervision for cross-view geo-localization, improving data quality through advanced curation strategies remains a promising direction for future work.

## 6 CONCLUSION

In this paper, we systematically address the challenges of CVGL through several key contributions. First, we provide a detailed analysis of the CVGL problem, highlighting the benefits of a unified framework for integrating multiple views and modalities, and emphasizing the importance of interpretability in practical applications. Second, we propose GLEAM-C, a foundational model that aligns diverse viewpoints and modalities with satellite imagery through a two-phase training strategy, achieving high accuracy and training efficiency. Third, we introduce GLEAM-X, a bilingual benchmark enabling explainable reasoning with human-interpretable explanations for image correspondences. GLEAM-C and GLEAM-X integrate into a unified pipeline combining accurate correspondence prediction with interpretable explanations, improving both robustness and trustworthiness of CVGL systems. We hope our work will advance research and development in this field.

ETHICS STATEMENT

This work relies exclusively on publicly available imagery, including satellite, UAV, panoramic, map, and ground-level data, and contains no personally identifiable or sensitive information. While geo-localization technologies may have dual-use risks, such as potential misuse in surveillance, our framework is intended solely for academic research, with transparency, interpretability, and fairness as primary goals. All code and data will be publicly released to support reproducibility, responsible research, and the safe advancement of explainable geo-localization.

REPRODUCIBILITY STATEMENT

We make extensive efforts to ensure the reproducibility of our work. We provide the main training and benchmarking code for review purposes in the Supplementary Materials. All code and datasets will be publicly released after the completion of the review process, ensuring that the experiments and results reported in this work can be fully reproduced by the community.

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

# A APPENDIX

## A.1 PSEUDOCODE FOR THE DDP LOSS COMPUTATION

We provide the pseudocode for DDP loss computation below.

---

**Algorithm 1:** Distributed InfoNCE Loss (DDP)

---

**Input:** Image features $\mathbf{f}_1, \mathbf{f}_2 \in \mathbb{R}^{B \times d}$, logit scale $\alpha$, world size $W$, rank $r$
**Output:** Contrastive loss $\mathcal{L}$ on rank $r$

1. Normalize features: $\mathbf{f}_1 \leftarrow \text{normalize}(\mathbf{f}_1)$, $\mathbf{f}_2 \leftarrow \text{normalize}(\mathbf{f}_2)$;
2. Gather features from all processes:

$$\{\mathbf{f}_1^{(w)}\}_{w=1}^{W} \leftarrow \text{AllGather}(\mathbf{f}_1), \quad \{\mathbf{f}_2^{(w)}\}_{w=1}^{W} \leftarrow \text{AllGather}(\mathbf{f}_2)$$

3. Concatenate features, excluding local rank duplicates:

$$\mathbf{F}_1 \leftarrow [\mathbf{f}_1] \cup \{\mathbf{f}_1^{(w)} \mid w \neq r\}, \quad \mathbf{F}_2 \leftarrow [\mathbf{f}_2] \cup \{\mathbf{f}_2^{(w)} \mid w \neq r\}$$

4. Compute logits:

$$\mathbf{Z}_1 = \alpha \cdot \mathbf{F}_1 \mathbf{F}_2^{\top}, \quad \mathbf{Z}_2 = \mathbf{Z}_1^{\top}$$

5. Construct global labels:

$$\mathbf{y} = [0, 1, \ldots, N-1], \quad N = \text{rows}(\mathbf{Z}_1)$$

6. Compute loss:

$$\mathcal{L} = \tfrac{1}{2}\big(\text{CE}(\mathbf{Z}_1, \mathbf{y}) + \text{CE}(\mathbf{Z}_2, \mathbf{y})\big)$$

**return** $\mathcal{L}$;

---

## A.2 DETAILS OF HUMAN ANNOTATION

To ensure the accuracy of the test set, we manually revise all test samples. Our test set consists of 504 query images, each paired with both a positive and a negative reference image, resulting in 1,008 image pairs. For each pair, we generate explanations in both Chinese and English, yielding 2,016 explanations in total.

We engage 4 human experts in Remote Sensing, each holding at least a master's degree, to perform data annotation and revision at $20 per hour per expert. We conduct two annotation rounds: in each round, experts verify the correctness of all four explanations (positive-Chinese, positive-English, negative-Chinese, negative-English) for each query image and correct them if necessary. If the experts cannot reach a consensus on whether a query and reference image match, we discard the query image and replace it with a suitable one from the remaining test data pool, then select new positive and negative reference images for it, followed by model-assisted generation and manual correction of the explanations. This ensures that all test samples are valid and unambiguous, providing a reliable basis for evaluation.

Among the 1,008 pairs (2,016 explanations), in Round 1, 596 explanations are directly modified and 68 pairs are replaced (34 query images, each with one positive and one negative pair); in Round 2, 48 explanations are directly modified and 8 pairs are replaced (4 query images, each with one positive and one negative pair). In total, 644 direct modifications are made to the explanations, and 76 pairs are replaced (38 query images, each with one positive and one negative pair) across both rounds. The decreasing number of modifications from Round 1 to Round 2 indicates progressive quality improvement and convergence toward high-quality pairs and explanations.

## A.3 LLM USAGE STATEMENT

We use Large Language Models (LLMs) solely to aid or polish writing in the preparation of this paper. Besides, commercial MLLMs are used to help construct the training and evaluation data (explainable reasoning annotations) for GLEAM-X. All core research activities—including experimental design, data analysis, and interpretation of results—are performed manually by the authors. LLMs do not contribute to research ideation or the generation of scientific conclusions, which remain entirely the work of the human authors.

## A.4 Deployment Analysis on Edge Devices

To evaluate the practical feasibility of our unified pipeline for real-world deployment, we conduct comprehensive performance profiling on an embedded platform. We deploy both GLEAM-C and GLEAM-X on an NVIDIA Jetson AGX Xavier (32GB RAM, JetPack 5.1) to evaluate real-world performance metrics. This analysis provides critical insights into the computational requirements and resource consumption in resource-constrained environments.

| Module | Model | Configuration | Peak Power (W) | Peak Memory (GB) | Time (s) |
|--------|-------|---------------|----------------|------------------|----------|
| GLEAM-C | ConvNeXt-B-384 | 1Q+10R | 30.4 | 5.4 | 4.3 |
| GLEAM-X | Qwen2.5-VL-3B | 1Q+1R | 35.7 | 12.1 | 28.5 |

Table A.1: Deployment performance on Jetson Xavier (Q: query, R: references).

These measurements demonstrate that our unified pipeline can run efficiently on edge devices with moderate computational resources. All measurements are obtained under full load conditions without quantization or optimization, indicating potential for further performance improvements in production deployments.

## A.5 Cross-Domain Generalization Analysis

We conduct cross-domain generalization experiments on ConvNeXt-B-384 to evaluate robustness across different views/modalities. We first pretrain the model on VIGOR only and test it on all four datasets without any fine-tuning. We then continue training on VIGOR+MAP to examine whether incremental learning improves generalization.

**Scenario 1: Zero-shot Transfer (VIGOR pretrain only).** Tab. A.2 shows the zero-shot transfer results, which reveal significant challenges in cross-modality generalization. When trained solely on VIGOR, the model achieves 76.60% R@1 on VIGOR but drops dramatically to 0.74% on MAP and 0.18% on SetVL. This substantial performance degradation stems from fundamental differences in viewpoint characteristics. These results indicate that features learned from one modality do not transfer well to drastically different view types without explicit training.

| Pretrain Dataset | Test Dataset | R@1 | R@5 | R@10 | Top-1 | AP/HR |
|------------------|--------------|-----|-----|------|-------|-------|
| VIGOR | VIGOR | 76.60 | 95.07 | 96.83 | 99.64 | 88.29 |
| | University-1652 | 19.93 | 37.37 | 46.59 | 47.89 | 24.43 |
| | MAP | 0.74 | 2.04 | 2.74 | 11.32 | 0.74 |
| | SetVL | 0.18 | 0.65 | 1.12 | 4.68 | 0.18 |

Table A.2: Zero-shot transfer performance.

**Scenario 2: Incremental Training (VIGOR pretrain → VIGOR + MAP fine-tune).** Table A.3 presents the incremental training results. Performance improves only for datasets included in training: MAP performance increases significantly from 0.74% to 93.85% R@1, while VIGOR maintains comparable performance (77.29% vs. 76.60%). However, unseen modalities experience negative transfer: University-1652 drops from 19.93% to 10.02%, and SetVL decreases from 0.18% to 0.04%. This suggests that simply adding one modality does not enable generalization to other unseen view types and may even harm performance on related but unseen modalities.

| Training Strategy | Test Dataset | R@1 | R@5 | R@10 | Top-1 | AP/HR |
|-------------------|--------------|-----|-----|------|-------|-------|
| VIGOR → VIGOR+MAP | VIGOR | 77.29 | 95.25 | 96.90 | 99.66 | 88.85 |
| | University-1652 | 10.02 | 20.80 | 27.67 | 28.78 | 13.12 |
| | MAP | 93.85 | 98.35 | 98.79 | 99.88 | 93.85 |
| | SetVL | 0.04 | 0.24 | 0.44 | 1.92 | 0.04 |

Table A.3: Incremental training performance.

**Comparison with Unified Multi-Dataset Training.** In contrast to the limited generalization observed in zero-shot and incremental scenarios, our unified framework with multi-dataset training (Tab. 2 in the main paper) demonstrates that when all modalities are jointly trained, the model achieves performance comparable to or better than single-dataset training. For instance, using PE-Core-L14-336 backbone, the two-phase training strategy achieves 75.96% R@1 on VIGOR (vs. 75.53% single-dataset), 93.97% on MAP (vs. 92.79%), and 23.25% on SetVL (vs. 21.34%). This validates that our architecture can effectively leverage diverse views and modalities when they are available during training, avoiding the negative transfer observed in incremental learning scenarios.

**Key Findings.** These findings highlight that while zero-shot cross-modality transfer remains challenging due to inherent viewpoint gaps, our unified approach successfully handles multiple modalities simultaneously when trained jointly. This demonstrates the importance of joint multi-dataset training for cross-view geo-localization, which is one of the primary contributions of our work.

## A.6 WHY VIGOR AS THE INITIAL DATASET?

We choose VIGOR as the initial dataset in our two-phase training strategy based on empirical evidence from preliminary experiments. This section focuses on the ConvNeXt-B-384 model to provide a more detailed justification.

**Empirical justification from Tab. 2.** As shown in Tab. 2 of the main paper, we compare three training strategies: (1) single-dataset training on each dataset separately, (2) from-scratch training on merged data from all datasets, and (3) two-phase training with VIGOR pre-training followed by merged-data training. The results reveal two critical findings:

**1)** Direct mixed-data training degrades VIGOR performance. When training from scratch on merged data, VIGOR Recall@1 drops to 73.37% compared to 76.60% in single-dataset training. This suggests that the model struggles to learn basic CVGL capabilities when simultaneously handling multiple datasets with different characteristics.

**2)** VIGOR pre-training enables effective multi-dataset learning. Using VIGOR as the initial phase, the two-phase strategy achieves 75.66% Recall@1 on VIGOR (recovering most of the performance loss) while significantly improving on other datasets: MAP increases from 92.52% to 94.05%, and SetVL-480K improves from 14.33% to 15.28%.

**Comparison with other pre-training datasets.** To validate the choice of VIGOR, we conduct additional experiments on ConvNeXt-B-384 to compare different initial datasets (VIGOR, University-1652, and SetVL-480K) before mixed-data training. Results are shown in Tab. A.4.

| Training Strategy | VIGOR R@1 | University R@1 | MAP R@1 | SetVL R@1 |
|---|---|---|---|---|
| **VIGOR → Merge (Tab. 2)** | 75.66 | 87.03 | 94.05 | 15.28 |
| **University → Merge** | 73.19 | 81.52 | 92.32 | 14.30 |
| **SetVL → Merge** | 70.55 | 75.64 | 91.93 | 14.39 |

Table A.4: Comparison of different initial datasets for two-phase training strategy.

Three key observations emerge from Tab. A.4:

**1)** VIGOR-first significantly outperforms all alternatives. It achieves +2.47% on VIGOR, +5.51% on University-1652, +1.73% on MAP, and +0.98% on SetVL compared to University-first pre-training, demonstrating its superior ability to provide effective initialization for multi-dataset learning.

**2)** University-first shows moderate but suboptimal performance. While it provides reasonable initialization, it significantly underperforms VIGOR-first, particularly on University-1652 itself (81.52% vs. 87.03%), suggesting it provides weaker cross-dataset transferability.

**3)** SetVL-first performs worst across all datasets. Despite being the largest dataset, SetVL's extreme difficulty (R@1 ∼14%) and highly diverse scene types result in poor performance on VIGOR (70.55%) and University-1652 (75.64%), making it unsuitable for initial feature learning.

**Why does VIGOR work better?** VIGOR provides superior initialization due to its *balanced difficulty*. As shown in Tab. 2, VIGOR achieves moderate performance in single-dataset training (R@1 76.60%), indicating it is neither too easy (like MAP at 92.60%) nor too difficult (like SetVL at 14.23%). This balanced difficulty allows the model to learn fundamental cross-view correspondence

reasoning without overfitting to trivial patterns or failing to converge on overly challenging scenarios. Consequently, VIGOR pre-training establishes a robust feature space that facilitates effective adaptation to diverse datasets in the subsequent mixed-data training phase.

## A.7 MORE SEMANTIC EVALUATION METHODS

Beyond Sentence-BERT, we introduce two additional evaluation methods to comprehensively assess the quality of generated explanations in the EN setting.

First, we conduct an LLM-as-a-judge evaluation using Gemini 2.5 Flash with a detailed 0-5 scoring rubric focusing on identifying geographical features and spatial reasoning accuracy. We choose Gemini 2.5 Flash because it is independent of the training data sources and less likely to show preference bias. Importantly, we provide Gemini with the query image, reference image, reference explanation, and model-generated explanation, enabling it to assess the factual correctness of the reasoning rather than just linguistic quality. Second, we perform a human evaluation using the same rubric with 4 human experts in Remote Sensing. To improve consistency in human evaluation, each expert evaluates one complete dataset among MAP, SetVL-480K, University-1652, and VIGOR. The detailed scoring rubric is provided in the Sec. A.11.

| Gemini 2.5 Flash | | | | | | | |
|---|---|---|---|---|---|---|---|
| **Model** | **0** | **1** | **2** | **3** | **4** | **5** | **Avg Score** |
| **GPT-4o** | 17.5% | 0.2% | 1.6% | 7.5% | 65.2% | 8.0% | 3.27 |
| **Doubao-1.5** | 19.5% | 0.0% | 1.0% | 4.1% | 66.0% | 9.4% | 3.25 |
| **Qwen2.5-VL-3B-Instruct** | 42.2% | 2.3% | 17.0% | 14.3% | 23.2% | 1.1% | 1.77 |
| **+label only** | 90.7% | 0.2% | 0.5% | 0.8% | 7.4% | 0.4% | 0.35 |
| **+GPT-4o** | 8.7% | 0.2% | 3.0% | 6.6% | 65.2% | 16.3% | 3.68 |
| **+Doubao-1.5** | 8.6% | 0.1% | 1.9% | 10.9% | 63.7% | 14.8% | 3.65 |

Table A.5: LLM-as-a-judge evaluation results with Gemini 2.5 Flash.

| Human Test | | | | | | | |
|---|---|---|---|---|---|---|---|
| **Model** | **0** | **1** | **2** | **3** | **4** | **5** | **Avg Score** |
| **GPT-4o** | 11.9% | 9.4% | 20.8% | 15.4% | 36.2% | 6.3% | 2.73 |
| **Doubao-1.5** | 13.4% | 10.4% | 15.7% | 12.4% | 40.2% | 7.9% | 2.79 |
| **Qwen2.5-VL-3B-Instruct** | 28.3% | 22.3% | 24.0% | 9.4% | 14.1% | 1.9% | 1.64 |
| **+label only** | 66.5% | 13.4% | 14.4% | 2.5% | 2.8% | 0.5% | 0.63 |
| **+GPT-4o** | 6.6% | 8.1% | 19.1% | 13.7% | 41.5% | 11.1% | 3.09 |
| **+Doubao-1.5** | 5.8% | 6.9% | 20.6% | 15.4% | 39.5% | 11.8% | 3.11 |

Table A.6: Human evaluation results with 4 experts.

The results are shown in Tab. A.5 and Tab. A.6. We observe consistent patterns across all three metrics. In the LLM-as-a-judge evaluation, the explanation-supervised models achieve average scores of 3.68 and 3.65, even surpassing their teacher models GPT-4o (3.27) and Doubao-1.5 (3.25), while substantially outperforming the baseline Qwen2.5-VL-3B-Instruct at 1.77. The human evaluation confirms this trend with scores of 3.09 and 3.11 versus 2.73 and 2.79 for the teachers and 1.64 for the baseline. The label-only model scores near zero (0.35 and 0.63) as it cannot generate explanations. These results align well with the Sentence-BERT scores in Tab. 5 of the main text, where the explanation-supervised models show similar improvements. The consistency across semantic similarity, LLM judgment, and human assessment demonstrates that the explanation-supervised models produce high-quality reasoning.

## A.8 HYPER-PARAMETERS

### A.8.1 GLEAM-C

Here we provide the training hyper-parameters of GLEAM-C using PE-Core-L14-336 ViT on the second training phase.

| Parameter | Value |
| --- | --- |
| Model | PE-Core-L14-336 |
| Epochs | 40 |
| Batch Size (Total) | 300 |
| Learning Rate | 1e-4 |
| Scheduler | Cosine |
| Warmup Epochs | 1 |
| Label Smoothing | 0.1 |
| Neighbour Select | 64 |
| Neighbour Range | 128 |
| Prob Rotate | 0.75 |
| Prob Flip | 0.5 |

Table A.7: Training hyper-parameters of GLEAM-C using PE-Core-L14-336.

### A.8.2 GLEAM-X

Here we provide the hyper-parameters of training Qwen2.5-VL-3B-Instruct on GLEAM-X.

| Parameter | Value |
| --- | --- |
| Learning Rate | 5e-5 |
| Batch Size (Total) | 60 |
| Epochs | 1 |
| Weight Decay | 0.1 |
| Warmup Ratio | 0.05 |
| Max Length | 2048 |
| Min/Max Pixels | 256/1296 |
| Gradient Checkpointing | True |
| Precision | BF16 |

Table A.8: Hyper-parameters of training Qwen2.5-VL-3B-Instruct on GLEAM-X.

## A.9 MORE DATA EXAMPLES OF GLEAM-X.

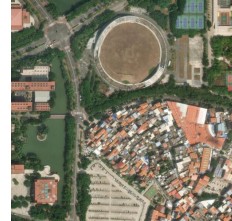

English explanation:

The two images depict the same area for a few reasons. Both images display a circular structure that appears to be a stadium or arena, which is centrally located. Surrounding this structure, the map and satellite images show a similar road layout, including a distinct curve on the western side and a rectangular area to the north. Additionally, the surrounding grid pattern of streets and nearby buildings match closely between the two images, including the presence of waterways and green spaces visible in both images.

Corresponding Chinese explanation:

这两张图显示的是同一地区，原因有几个。两幅图中都有一个中央的圆形建筑，看起来像是体育场或竞技场。围绕这个结构，地图和卫星图显示了相似的道路布局，包括西侧明显的弯曲和北侧的矩形区域。此外，周围街道和附近建筑的网格模式在两张图中非常相似，图中还可以看到水道和绿色空间。

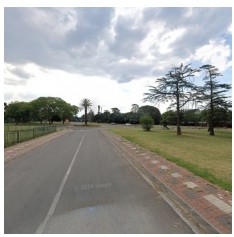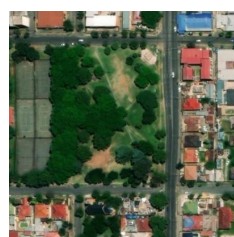

English explanation:

The two images do not correspond to the same area due to several noticeable differences. In the first image, taken from the ground level, there is a roundabout with a road that has a marked bike path and a variety of trees, including a prominent single palm tree in the middle of the roundabout. It shows a wide, open space with organized urban landscaping. In contrast, the second image is a satellite view showing a rectangular park surrounded by a grid of streets. This park has tennis courts and dense clusters of trees, which are not evident in the first image. The layout and elements present in both images indicate they depict different locations.

Corresponding Chinese explanation:

这两张图片不对应于同一个地区，因为有几个显著的区别。第一张地面图片显示了一个环形交叉路口，道路上有自行车道标志，还有各种树木，包括一个显眼的单棵棕榈树。它展示了一个宽阔的开放空间和有组织的城市绿化。而第二张卫星图显示的是一个被街道网格包围的矩形公园。这个公园有网球场和密集的树丛，这些在第一张图片中没有出现。图片的布局和元素显示它们描述的是不同地点。

Figure 3: Street map and ground photograph samples.

English explanation:

The first image is a panoramic view from a street-level perspective, showing a street scene under an elevated train track structure. The recognizable feature is the steel structure supporting the train tracks above, with cars parked on the street below. The second image is a satellite view showing an urban area with an elevated track visible as a thin line crossing over a street. The satellite image clearly shows the elevated structure aligned with the street below, which matches the street structure and alignment shown in the panoramic view. The presence of buildings and the layout of the road network support the idea that both images represent the same location.

Corresponding Chinese explanation:

第一张图片是从街道层面的全景视角，展示了高架铁路轨道结构下的街景。可以辨认出的特征是上方的火车轨道支撑的钢结构，下面是街道上停放的汽车。第二张图片是卫星视图，显示一个城市区域，有一个明显的高架轨道如同一条细线横跨街道。卫星图像清楚地显示高架结构与下面的街道对齐，这与全景图中展示的街道结构和对齐方式一致。建筑物的存在和道路网络的布局支持了这两个图像代表同一位置的观点。

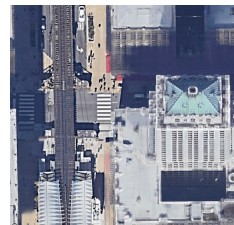

English explanation:

The two images do not correspond to the same area for several reasons. First, the architectural style of the buildings is different; the first image features a more modern structure with clear geometric lines and distinct surfaces, while the second image shows a complex of buildings with different layouts. Second, the surrounding environment is not similar: the first image has dense greenery and natural surroundings, whereas the second image has an organized path layout. Additionally, the second image shows more connections between buildings, indicating a different setting, possibly a campus or a similar complex.

Corresponding Chinese explanation:

这两张图片对应的区域不同，原因有几个。首先，建筑风格不同；第一张图中的建筑呈现出现代的风格，有明显的几何线条和表面，而第二张图中的建筑群布局不同。其次，周围环境不同：第一张图展示了密集的绿地和自然环境，而第二张图则有井然有序的小路布局。此外，第二张图片中的建筑物之间有更多的连接，表明这是一个不同的环境，可能是一个校园或类似的建筑群。

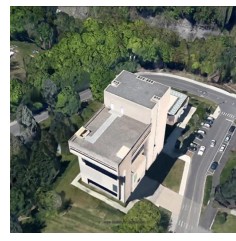 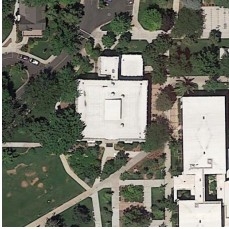

Figure 4: Panoramic view and UAV imagery samples.

## A.10 MORE QUALITATIVE RESULTS ON GLEAM-X BENCHMARK

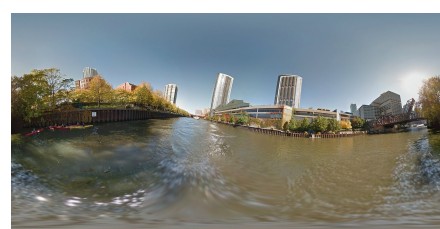 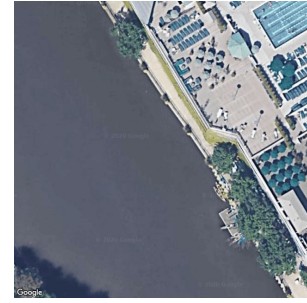

**Ground-Truth**

[[1]]\n\n这两幅图像很可能对应同一地点，因为两者中都可见明显的特征。在全景图中，河流和周围的建筑物显示出一个独特的城市滨水区。卫星图像显示了一个有相邻建筑物和游泳池的河岸，这表明它也是一个城市河边位置。两幅图像中常见的相似结构，如高层建筑和沿河的树木、路径，都支持它们描绘的是同一地点的观点。

[[1]]\n\nThe two images likely correspond to the same area due to identifiable features visible in both. In the panoramic image, the river and surrounding buildings suggest a distinctive urban waterfront area. The satellite image shows a waterfront with adjacent buildings and a swimming pool, indicating it's also an urban riverside location. The presence of similar structures, like high-rise buildings and waterfront features, such as trees lining the river and pathways, is common in both images, supporting the idea that they depict the same location.

**GPT-4o**

[[1]]\n\n这两张图片看起来对应同一个地区，主要理由如下：\n\n1. **水体和建筑：** 第一张全景图显示了一个城市河流的景观，周围有现代化的高楼建筑。卫星图也显示了一个水体周围有建筑的区域。整体城市和水体布局相似。\n\n2. **环境特征：** 全景图中可以看到河流和沿河绿化，卫星图中也明显有沿水体的绿地和路径。\n\n3. **桥梁和建筑布局：** 从全景图中可以看到桥梁的结构和建筑的分布，似乎与卫星图中建筑和道路的配置相吻合。\n\n综合以上因素，两个图像展示的环境特征和建筑特点相似，很可能是同一个地区。

[[1]]\n\nThese two images appear to correspond to the same area, primarily for the following reasons:\n\n1. **Water bodies and buildings:** The first panorama depicts a city river landscape surrounded by modern high-rise buildings. The satellite image also shows an area with buildings surrounding a water body. The overall urban and water body layout is similar.\n\n2. **Environmental features:** The panorama reveals a river with riverside greenery, while the satellite image clearly shows green spaces and pathways along the water body.\n\n3. **Bridge and Building Layout:** The panorama depicts bridge structures and building distributions that appear to align with the configuration of buildings and roads in the satellite image.\n\nConsidering these factors collectively, the environmental features and architectural characteristics displayed in both images are similar, strongly suggesting they depict the same area.

**Qwen2.5-VL-3B-Instruct**

[[0]]\n\n解释：\n两张图片虽然都展示了河流和城市景观，但它们对应的是不同的地区。第一张图片显示的是一个城市河岸，有建筑物、树木和船只，背景是蓝天。第二张图片显示的是一个湖泊或河流的卫星图像，有码头、船只和树木，但没有建筑物。因此，这两张图片对应的是不同的地区。

[[0]]\n\nAlthough both images depict rivers and urban landscapes, they correspond to different regions. The first image shows a city riverbank with buildings, trees, and boats against a blue sky backdrop. The second image presents a satellite view of a lake or river featuring docks, boats, and trees, but no buildings. Therefore, these two images correspond to distinct locations.

**Qwen2.5-VL-3B-Instruct + label only**

[[1]]

**Qwen2.5-VL-3B-Instruct + GPT-4o**

[[1]]\n\n这两张图片对应同一地区，因为它们都显示了一个滨水区域，有明显的特征。在全景图中，可以看到一条河流或运河，河岸边有木制的护岸，旁边有建筑物和树木。卫星图显示了类似的滨水区域，有水体、岸边的树木和建筑物。卫星图中建筑物的形状和布局与全景图中看到的建筑相匹配。此外，全景图中可见的桥梁结构与卫星图中水边的结构相对应。这些共同的元素确认了两张图片是同一地点的不同视角。

[[1]]\n\nThese two images correspond to the same area, as both depict a distinct waterfront region with recognizable features. The panoramic image shows a river or canal with wooden revetments along the banks, flanked by buildings and trees. The satellite image reveals a similar waterfront area featuring a body of water, trees along the shoreline, and structures. The shapes and layout of the buildings in the satellite image match those visible in the panoramic view. Additionally, the bridge structure discernible in the panoramic view corresponds to the structure near the water's edge in the satellite image. These shared elements confirm that the two images represent different perspectives of the same location.

Figure 5: Sample on the VIGOR test set (Chinese scenario). The gray English text is a direct translation of the Chinese response.

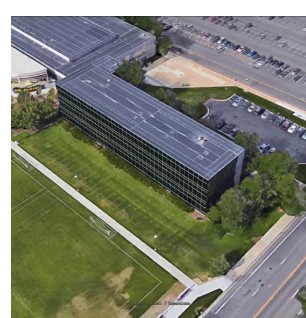 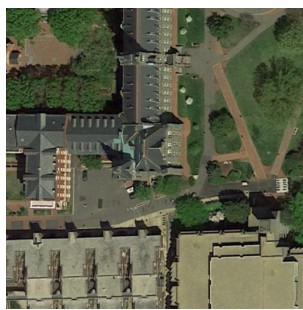

**Ground-Truth**

[[0]]\n\nThe two images do not correspond to the same area for several reasons. In the first image, there is a building with a modern architectural style, characterized by flat roofs and large panes of reflective glass, typical of an office or technology park. It is surrounded by parking lots and green space, including a soccer field. In contrast, the second image shows buildings with classic architecture, featuring more traditional designs with pointed roofs and a courtyard area, which are often associated with historical institutions such as universities or government buildings. Additionally, the layout of pathways and the arrangement of the surrounding greenery differ significantly between the two images.

**Doubao-1.5-Thinking-Vision-Pro**

[[0]]\n\nExplanation: To determine if the two images correspond to the same area, we analyze key features: \n1. **Building Architecture and Roof**: The first image shows a building with a dark - colored roof (likely solar panels or dark roofing material) and extensive glass - walled facades. The second image features a building with a light - colored, flat roof and a large rectangular skylight. These architectural differences are significant. \n2. **Surrounding Landscape and Layout**: The first image has a soccer field with visible goalposts adjacent to the building, along with parking lots and multi - lane roads. The second image shows a different field layout (no visible goalposts matching the first image's field) and a distinct road and sidewalk arrangement. \n3. **Contextual Elements**: The overall environment, including tree placement, road patterns, and the relationship between buildings and open spaces, does not match between the two images. \n\nDue to these substantial differences in building design, landscape features, and spatial context, the two images do not correspond to the same area.

**Qwen2.5-VL-3B-Instruct**

[[0]]\n\nExplanation:\nThe two images provided do not correspond to the same area. The first image is an aerial view of a building with a flat roof and a parking lot in the background. The second image is a satellite view of a different area, showing a building with a sloped roof and a grassy area in the foreground. The buildings and their surroundings in the two images are distinct, indicating that they are not the same location.

**Qwen2.5-VL-3B-Instruct + label only**

[[0]]

**Qwen2.5-VL-3B-Instruct + Doubao-1.5-Thinking-Vision-Pro**

[[0]]\n\nThe first image shows a modern building with a large glass facade, solar panels on the roof, and a sports field adjacent to it, along with a parking lot. The second image depicts a more traditional, possibly institutional or historical building complex with multiple connected structures, courtyards, and a different layout of green spaces and pathways. The architectural styles, building layouts, and surrounding features (like the sports field vs. the courtyard) are distinct, indicating different areas.

Figure 6: Sample on the University-1652 test set (English scenario).

## A.11 EVALUATION RUBRIC FOR LLM-AS-A-JUDGE AND HUMAN EVALUATION

```
judge_en_prompt = '''
# Task Introduction

This task aims to evaluate the quality of explanations for cross-view geo-localization. Given
two images (which may include satellite imagery, UAV imagery, street maps, panoramic images,
and ground photos), annotators must determine whether they depict the same geographical
location and provide a detailed explanation for their judgment. The explanation should
identify specific geographical features and demonstrate spatial reasoning to support the
matching decision. You are now provided with two images to evaluate, a model answer, and a
reference answer. Please score the model answer strictly according to the following scoring
criteria (0-5 points) and provide justification for your score.

# Scoring Criteria

## 5 Points - Excellent
The explanation quality significantly exceeds the reference annotation, identifying more
correct specific geographic features, or demonstrating markedly superior precision and depth
in spatial reasoning compared to the reference annotation. The argumentation is logically
rigorous, feature descriptions are highly verifiable and distinctive, with no factual errors.
Demonstrates exceptional spatial analysis capabilities and can provide equally valid
argumentative dimensions not covered by the reference annotation.

Example: [[1]]\n\nBoth images show the same area in Munich: An oval-shaped stadium is visible
in the center of the satellite image, labeled as Olympiastadion on the map; the curved lake
shape on the west side of the stadium matches perfectly; the spacing and orientation of three
parallel roads on the north side are consistent; the rectangular parking lot position in the
southeast corner corresponds; additionally, a characteristic spiral ramp structure is visible
on the south side of the stadium, which precisely matches the building outline on the map.

## 4 Points - Good (Reference annotations typically correspond to 4-point level)
The explanation identifies geographic features that are essentially consistent with the
reference annotation or other correct related geographic features, with accurate spatial
reasoning, coherent logic, and no critical errors. Can adequately explain the main basis for
match or non-match, reaching the quality level of the reference annotation. Directional
descriptions or feature positioning may be somewhat general, but this does not affect the
validity of the argumentation.

Example: [[0]]\n\nThe two images do not correspond: The map shows a regular grid-pattern block
 with obvious roundabouts and radial roads, while the satellite image shows irregular curved
roads surrounded by large areas of farmland. The road patterns and land use types are clearly
different.

## 3 Points - Acceptable
The explanation identifies some correct features but lacks specificity in description, spatial
 analysis is relatively superficial, and argumentation is insufficient. Although the
conclusion is correct, the reasoning process has vagueness or generalization issues. There may
 be minor factual errors that do not affect the final judgment, such as slight misjudgment of
feature types.

Example: [[1]]\n\nThe two images match because both show rivers and bridges. The water body
shape in the satellite image is consistent with the river seen in the panoramic image, and
both are surrounded by buildings.

## 2 Points - Poor
Although the match judgment is correct, the explanation contains critical factual errors,
spatial logic fallacies, or identified features are overly generalized and lack
distinctiveness. For non-match cases, may misjudge image modality differences (perspective,
color, resolution) as geographic feature differences. The reasoning lacks validity and is
difficult to provide substantial support for the judgment.

Example: [[0]]\n\nNo match, because the buildings in the first image have red roofs, while the
 building roofs in the second satellite image are gray. The different colors indicate
different places.

## 1 Point - Very Poor
Although the match judgment is correct, the reasoning process has fundamental flaws, including
 hallucinating non-existent features, logical confusion, or using task-irrelevant factors (
image quality, shooting conditions, weather conditions) as primary arguments. Demonstrates
serious misunderstanding of the task objective.

Example: [[0]]\n\nDoes not correspond, because the first is a panoramic image taken during the
 day, and the second is a satellite image, with different shooting methods. Moreover, the
first image is clearer, while the second is more blurry.

## 0 Points - Fail
The match judgment is incorrect, or no valid explanation is provided (blank, label only, off-
topic).
```

```
---

# Evaluation Dimensions

**Feature Accuracy:** Whether the identified geographic features actually exist. Can be
consistent with the reference annotation or provide other valid correct features. Avoid
feature hallucination or misidentification.

**Reasoning Validity:** Whether the argumentation adequately supports the match judgment. For
match cases, need to demonstrate consistency of spatial features; for non-match cases, need to
 demonstrate essential differences in geographic features rather than image modality
differences.

**Description Specificity:** Whether feature descriptions are distinctive and localizable.
High-quality explanations should include specific spatial positioning information, rather than
 generalized descriptions applicable to any location.

**Verifiability:** All statements should be verifiable through image observation, avoiding
subjective speculation or unverifiable assertions.

---

# Output Format (Must Strictly Follow)

Format: [[<score>]]\n\n<brief explanation>

- <score>: Integer between 0-5
- <brief explanation>: One sentence explaining the scoring rationale

**Output Examples:**

[[5]]\n\nIdentified 7 precisely matching geographic features with accurate spatial positioning
 and rigorous argumentation.

[[3]]\n\nConclusion is correct but feature descriptions are too general, lacking specific
spatial positioning information.

[[0]]\n\nMatch judgment is incorrect.
'''
```

