# OpenReview forum: "GLEAM: Learning to Match and Explain in Cross-View Geo-Localization"
_ICLR.cc/2026/Conference — ICLR 2026 Conference Withdrawn Submission_

### Official Review · Reviewer_kFbq · 2025-10-30

**Soundness:** 2
**Presentation:** 3
**Contribution:** 2
**Rating:** 4
**Confidence:** 4

**Summary:**

This paper proposes a unified and interpretable framework for cross-view geo-localization. The first module, GLEAM-C, aligns multiple modalities (UAV, map, panorama, ground) with satellite images through contrastive learning and a two-phase training strategy, achieving strong accuracy and 5× faster training via distributed data parallelism. The second module, GLEAM-X, introduces an explainable reasoning benchmark where multimodal large language models (e.g., Qwen2.5-VL) generate bilingual (English–Chinese) natural-language explanations for image correspondences, using data annotated by GPT-4o and Doubao-1.5. Together, they form an integrated system that enhances both performance and interpretability in geo-localization tasks

**Strengths:**

1. GLEAM-C integrates diverse modalities—UAV, street maps, panoramas, and ground images—into a single contrastive learning framework aligned with satellite imagery. This unified design eliminates the need for modality-specific architectures and enhances generalization across heterogeneous views.

2.The model adopts a two-phase training procedure (VIGOR pretraining followed by multi-dataset fusion) combined with a Distributed Data Parallel implementation that aggregates cross-GPU features. This innovation accelerates training by more than fivefold while maintaining accuracy, enabling large-scale foundational CVGL learning.

3. GLEAM-X extends CVGL beyond retrieval to interpretable reasoning, coupling cross-view matching with bilingual (English–Chinese) explanation generation using multimodal large language models. Fine-tuning Qwen2.5-VL on high-quality annotated data improves both matching accuracy and transparency over commercial LLMs.

**Weaknesses:**

There are several concerns that I would like the authors to address.

First, the paper does not clearly justify why all modalities were aligned exclusively with satellite imagery. It would be important to provide reasoning or an ablation to show why other modalities were not used as reference anchors.

Second, the work appears largely as an engineered combination of existing ideas rather than a conceptually novel framework. For instance, the core methodology draws heavily from established approaches such as InfoNCE-based contrastive learning and OpenCLIP-style distributed training, without introducing a distinct algorithmic innovation.

Third, the evaluation of explanation quality raises validity concerns — since the reference explanations are GPT-generated and only the test set is human-edited, the model is effectively rewarded for mimicking its teacher rather than for producing truly accurate or human-intelligible reasoning.

Additionally, Section 3.3.1 lacks justification for choosing VIGOR as the initial dataset in the two-phase training strategy.
Finally, no information is provided on inter-annotator agreement during the human revision of the explanations — including how many samples were discarded, or qualitative feedback from annotators on the outputs of the multimodal LLMs.

**Questions:**

I would request the authors to comment on the points mentioned in the weaknesses part. Specifically,


What is the rationale for aligning all modalities exclusively with satellite imagery? Have the authors explored or analyzed alternative anchor modalities, or can they provide evidence (empirical or conceptual) supporting this design choice?


Beyond combining existing components such as InfoNCE loss and OpenCLIP-style DDP, what specific algorithmic or architectural innovation does GLEAM introduce? How does it advance the state of the art beyond prior CVGL frameworks?


Since GPT-generated explanations serve as the primary supervision and evaluation reference, how do the authors ensure that the model learns genuine interpretive reasoning rather than merely imitating GPT-style phrasing? Were any human-centric or factual quality metrics used?


Why was VIGOR chosen as the initial dataset in the two-phase training process? Have the authors verified that starting with VIGOR (instead of other datasets) leads to measurable benefits in generalization or convergence?


Could the authors report inter-annotator agreement scores or qualitative statistics (e.g., number of samples revised or rejected) to substantiate the reliability of the human-refined explanation dataset?

---

> ### Author Response · Authors · 2025-11-16
>
> We appreciate your thorough analysis and critical questions.
>
> >***Weaknesses 1: First, the paper does not clearly justify why all modalities were aligned exclusively with satellite imagery. It would be important to provide reasoning or an ablation to show why other modalities were not used as reference anchors.***
>
> Thank you for this important question. We would like to clarify that using satellite imagery as the reference anchor follows the definition of cross-view geo-localization (CVGL), not an arbitrary design choice requiring ablation studies. As defined in recent literature, CVGL generally aims to determine the localization of query images by retrieving the most similar GPS-tagged satellite reference images [1,2,3]. Satellite images are orthorectified products that are strictly aligned with geographic coordinates and geospatial information, providing comprehensive Earth surface coverage with precise location coordinates. This makes them the natural geo-tagged reference for geographic localization tasks.
>
> Using other modalities as reference anchors would be impractical for several reasons. Ground, UAV, or panoramic imagery lack comprehensive geographic coverage and do not have inherent geo-alignment properties. They represent query modalities rather than reference coordinates. While street maps exist, they are abstract vector representations that lack actual ground surface information such as building appearances, terrain textures, and vegetation, which are necessary for visual matching in CVGL. Only satellite imagery combines both precise geo-tagging and rich visual ground surface information. Therefore, aligning all modalities with satellite imagery follows the fundamental principle of CVGL. Alternative designs would contradict the task definition itself and thus do not require ablation studies.
>
> [1] Chen Z, Yang Z X, Rong H J. Multi-level embedding and alignment network with consistency and invariance learning for cross-view geo-localization[J]. IEEE Transactions on Geoscience and Remote Sensing, 2025.
>
> [2] Li G, Qian M, Xia G S. Unleashing unlabeled data: A paradigm for cross-view geo-localization[C]//Proceedings of the IEEE/CVF Conference on Computer Vision and Pattern Recognition. 2024: 16719-16729.
>
> [3] Xia P, Wan Y, Zheng Z, et al. Enhancing cross-view geo-localization with domain alignment and scene consistency[J]. IEEE Transactions on Circuits and Systems for Video Technology, 2024.

---

> > ### Author Response · Authors · 2025-11-16
> >
> > >***Weaknesses 2: Second, the work appears largely as an engineered combination of existing ideas rather than a conceptually novel framework. For instance, the core methodology draws heavily from established approaches such as InfoNCE-based contrastive learning and OpenCLIP-style distributed training, without introducing a distinct algorithmic innovation.***
> >
> > We respectfully disagree with the characterization of our work as merely an "engineered combination." Innovation in research isn't limited to inventing new algorithms or architectures. It equally involves problem formulation, system design, and empirical validation, all of which are central to our contributions. In fact, many highly-regarded papers in top venues use established tools like contrastive learning, transformers, or diffusion models to tackle new problems. What matters is not whether we invent entirely new algorithmic primitives, but how we design frameworks, formulate problems, and validate solutions to address real challenges.
> >
> > Our work makes significant contributions in task formulation. GLEAM-C is the first unified multi-view/modal CVGL framework that handles ground-level, UAV, panoramic images, and street maps within a single model to localize against satellite imagery. Prior work focuses on single query views or modalities (e.g., ground-to-satellite only) and requires training separate models for each view/modality, which is both inefficient and impractical for real-world deployment. GLEAM-X goes further by integrating explainable reasoning into CVGL for the first time, introducing a paradigm that combines correspondence prediction with human-interpretable explanations. These task definitions are contributions in their own right.
> >
> > Methodologically, GLEAM-C introduces several key innovations. We propose a two-phase progressive training strategy that begins with single-modality training before transitioning to multi-modal learning. We also leverage distributed training with DDP to handle large-scale multi-modal data efficiently. For GLEAM-X, we construct a high-quality bilingual (Chinese-English) benchmark dataset for correspondence reasoning and fine-tune Qwen2.5-VL-3B to generate human-understandable explanations, bridging the gap between retrieval accuracy and model transparency.
> >
> > The system integration itself represents another layer of contribution. GLEAM-C handles localization while GLEAM-X provides verification and explainability. Together, they form a complete end-to-end interpretable CVGL system that enhances both robustness and user trust. This integrated pipeline addresses practical needs in UAV navigation, emergency response, and urban planning, where both accuracy and interpretability matter.
> >
> > It's worth clarifying that tools are not the same as methods. InfoNCE and OpenCLIP are tools, just like CNNs or transformers. The real question is how we design the overall framework, devise training strategies, and integrate components to solve new problems. Thoughtful engineering and system design are valuable research contributions, not just "engineering."
> >
> > Our empirical results validate this stance. The proposed methods demonstrate competitive or superior performance against baselines and state-of-the-art approaches on both existing benchmarks and newly introduced datasets. These results don't come from simple combination. They reflect careful design choices and systematic exploration. Our work addresses real-world challenges with demonstrated practical impact, combining scientific rigor with engineering value.

---

> > > ### Author Response · Authors · 2025-11-16
> > >
> > > >***Weaknesses 3: Third, the evaluation of explanation quality raises validity concerns — since the reference explanations are GPT-generated and only the test set is human-edited, the model is effectively rewarded for mimicking its teacher rather than for producing truly accurate or human-intelligible reasoning.***
> > >
> > > #### 1. How to ensure that the model learns genuine interpretive reasoning rather than merely imitating GPT-style phrasing
> > >
> > > Thank you for raising this important validity concern. We acknowledge that using GPT-generated references could risk rewarding stylistic mimicry rather than genuine reasoning quality. Our approach addresses this issue through several key design choices.
> > >
> > > First, ground-truth guidance during data generation directly mitigates the risk of learning incorrect reasoning patterns. When generating explanations, we provide GT matching labels to teacher models, steering them toward correct correspondence reasoning from the start. This ensures that the generated explanations are grounded in factually correct matches rather than plausible-sounding but incorrect reasoning.
> > >
> > > Second, our human-edited test set serves as an independent validation benchmark. Tab. 4-5 show that teacher models (GPT-4o and Doubao-1.5) achieve 77-84% average accuracy and 0.74-0.83 average similarity when evaluated on this human-curated test set without GT labels during inference. In contrast, student models (Qwen2.5-VL-3B) trained on GT-guided explanations achieve 87-91% accuracy and 0.79-0.84 similarity on the same human-edited test set. The fact that student models surpass their teachers on human-validated data demonstrates that the training process captures genuine reasoning capabilities rather than merely replicating teacher artifacts.
> > >
> > > We acknowledge that explanation quality ultimately depends on teacher model capabilities, which set an upper bound on reasoning depth. As this is the first work to introduce explanation supervision for cross-view geo-localization, we view our current approach as establishing a strong baseline. Future work could explore iterative refinement with human feedback, multi-teacher distillation, or adversarial filtering to further improve explanation quality. However, the current results demonstrate that GT-guided generation combined with human-validated evaluation provides a solid foundation for learning interpretable correspondence reasoning.

---

> > > > ### Author Response · Authors · 2025-11-16
> > > >
> > > > #### 2. Adding human-centric or factual quality metrics
> > > >
> > > > We add two additional evaluation methods (EN) to address your concern.
> > > >
> > > > First, we conduct an LLM-as-a-judge evaluation using Gemini 2.5 Flash with a detailed 0-5 scoring rubric focusing on identifying geographical features and spatial reasoning accuracy. We choose Gemini 2.5 Flash because it is independent from the training data sources and less likely to show preference bias. Importantly, we provide Gemini with the query image, reference image, reference explanation, and model-generated explanation, enabling it to assess factual correctness of the reasoning rather than just linguistic quality. Second, we perform a human evaluation using the same rubric with 4 human experts in Remote Sensing. To improve consistency in human evaluation, each expert evaluates one complete dataset among MAP, SetVL-480K, University-1652, and VIGOR. The detailed scoring rubric is provided in the Appendix.
> > > >
> > > > The results below show consistent patterns across all three metrics. In the LLM-as-a-judge evaluation, the explanation-supervised models achieve average scores of 3.68 and 3.65, even surpassing their teacher models GPT-4o (3.27) and Doubao-1.5 (3.25), while substantially outperforming the baseline Qwen2.5-VL-3B-Instruct at 1.77. The human evaluation confirms this trend with scores of 3.09 and 3.11 versus 2.73 and 2.79 for the teachers and 1.64 for the baseline. The label-only model scores near zero (0.35 and 0.63) as it cannot generate explanations. These results align well with the Sentence-BERT scores in Tab. 5 of the paper where the explanation-supervised models show similar improvements. The consistency across semantic similarity, LLM judgment, and human assessment demonstrates that the explanation-supervised models produce high-quality reasoning.
> > > >
> > > >
> > > > | Gemini   2.5 Flash |  |  |  |  |  |  |  |
> > > > |---|:---:|:---:|:---:|:---:|:---:|:---:|:---:|
> > > > | Model | 0 | 1 | 2 | 3 | 4 | 5 | Avg Score |
> > > > | GPT-4o | 17.5% | 0.2% | 1.6% | 7.5% | 65.2% | 8.0% | 3.27  |
> > > > | Doubao-1.5 | 19.5% | 0.0% | 1.0% | 4.1% | 66.0% | 9.4% | 3.25  |
> > > > | Qwen2.5-VL-3B-Instruct | 42.2% | 2.3% | 17.0% | 14.3% | 23.2% | 1.1% | 1.77  |
> > > > | Qwen2.5-VL-3B+label only | 90.7% | 0.2% | 0.5% | 0.8% | 7.4% | 0.4% | 0.35  |
> > > > | Qwen2.5-VL-3B+GPT-4o | 8.7% | 0.2% | 3.0% | 6.6% | 65.2% | 16.3% | 3.68  |
> > > > | Qwen2.5-VL-3B+Doubao-1.5 | 8.6% | 0.1% | 1.9% | 10.9% | 63.7% | 14.8% | 3.65  |
> > > >
> > > >
> > > > | Human Test |  |  |  |  |  |  |  |
> > > > |---|:---:|:---:|:---:|:---:|:---:|:---:|:---:|
> > > > | Model | 0 | 1 | 2 | 3 | 4 | 5 | Avg Score |
> > > > | GPT-4o | 11.9% | 9.4% | 20.8% | 15.4% | 36.2% | 6.3% | 2.73  |
> > > > | Doubao-1.5 | 13.4% | 10.4% | 15.7% | 12.4% | 40.2% | 7.9% | 2.79  |
> > > > | Qwen2.5-VL-3B-Instruct | 28.3% | 22.3% | 24.0% | 9.4% | 14.1% | 1.9% | 1.64  |
> > > > | Qwen2.5-VL-3B+label only | 66.5% | 13.4% | 14.4% | 2.5% | 2.8% | 0.5% | 0.63  |
> > > > | Qwen2.5-VL-3B+GPT-4o | 6.6% | 8.1% | 19.1% | 13.7% | 41.5% | 11.1% | 3.09  |
> > > > | Qwen2.5-VL-3B+Doubao-1.5 | 5.8% | 6.9% | 20.6% | 15.4% | 39.5% | 11.8% | 3.11  |

---

> > > > > ### Author Response · Authors · 2025-11-16
> > > > >
> > > > > >***Weaknesses 4.1: Additionally, Section 3.3.1 lacks justification for choosing VIGOR as the initial dataset in the two-phase training strategy.***
> > > > >
> > > > > Thank you for this question. We choose VIGOR as the initial dataset in the two-phase training strategy based on empirical evidence from preliminary experiments. In our explanation, we focus on the ConvNeXt-B-384 model.
> > > > >
> > > > > #### 1. Empirical justification from Tab. 2 of the paper
> > > > >
> > > > > As shown in Table 2 (ConvNeXt-B-384) of our paper, we compare three training strategies:
> > > > > - Single-dataset training: Training separately on each dataset
> > > > > - From-scratch (Merge): Directly training on mixed data from all datasets
> > > > > - Two-phase (VIGOR → Merge): Pre-training on VIGOR, then training on mixed data
> > > > >
> > > > > The results reveal two critical findings:
> > > > >
> > > > > Finding 1: Direct mixed-data training degrades VIGOR performance. When training from scratch on merged data, VIGOR Recall@1 drops to 73.37% compared to 76.60% in single-dataset training. This suggests that the model struggles to learn basic CVGL capabilities when simultaneously handling multiple datasets with different characteristics.
> > > > >
> > > > > Finding 2: VIGOR pre-training enables effective multi-dataset learning. Using VIGOR as the initial phase, the two-phase strategy achieves 75.66% Recall@1 on VIGOR (recovering most of the performance loss) while significantly improving on other datasets: MAP increases from 92.52% to 94.05%, and SetVL-480K improves from 14.33% to 15.28%.
> > > > >
> > > > > #### 2. Why VIGOR specifically?
> > > > >
> > > > > We conduct extra experiments (ConvNeXt-B-384) comparing different pre-training datasets (VIGOR, University-1652, and SetVL-480K) before mixed-data training:
> > > > >
> > > > > | Training Strategy | VIGOR R@1 | University R@1 | MAP R@1 | SetVL R@1 |
> > > > > |---------------------|-----------|----------------|---------|-----------|
> > > > > | VIGOR → Merge (Tab. 2) | 75.66 | 87.03 | 94.05 | 15.28 |
> > > > > | University → Merge | 73.19 | 81.52 | 92.32 | 14.30 |
> > > > > | SetVL → Merge | 70.55 | 75.64 | 91.93 | 14.39 |
> > > > >
> > > > > Key observations:
> > > > >
> > > > > (1) VIGOR-first significantly outperforms all other strategies across all datasets: +2.47% on VIGOR, +5.51% on University-1652, +1.73% on MAP, and +0.98% on SetVL compared to University-first pre-training.
> > > > >
> > > > > (2) University-first shows moderate performance but significantly underperforms VIGOR-first, particularly on University-1652 itself (81.52% vs. 87.03%), suggesting it provides weaker initialization for multi-dataset learning.
> > > > >
> > > > > (3) SetVL-first performs worst across all datasets, with particularly poor results on VIGOR (70.55%) and University-1652 (75.64%). Despite being the largest dataset, SetVL's extreme difficulty (R@1 ~14%) and highly diverse scene types make it unsuitable for initial feature learning.
> > > > >
> > > > > #### 3. Why does VIGOR work better?
> > > > >
> > > > > VIGOR provides superior initialization for the reason of balanced difficulty: VIGOR achieves moderate performance in single-dataset training (R@1 76.60%), indicating it is neither too easy (like MAP at 92.60%) nor too difficult (like SetVL at 14.23%). This balanced difficulty allows the model to learn fundamental cross-view correspondence reasoning without overfitting to trivial patterns or failing to converge.

---

> > > > > > ### Author Response · Authors · 2025-11-16
> > > > > >
> > > > > > >***Weaknesses 4.2: Finally, no information is provided on inter-annotator agreement during the human revision of the explanations — including how many samples were discarded, or qualitative feedback from annotators on the outputs of the multimodal LLMs.***
> > > > > >
> > > > > > Thank you for pointing out this important detail. To ensure the accuracy of the test set, we manually revise all test samples. Our test set consists of 504 query images, each paired with both a positive and a negative reference image, resulting in 1,008 image pairs. For each pair, we generate explanations in both Chinese and English simultaneously in a single prompt, explicitly asking the model to provide corresponding descriptions in both languages. This approach ensures semantic alignment in the bilingual dataset, yielding 2,016 explanations in total.
> > > > > >
> > > > > > We engage 4 human experts in Remote Sensing, each holding at least a master's degree, to perform data annotation and revision. We conduct two annotation rounds: in each round, experts verify the correctness of all four explanations (positive-Chinese, positive-English, negative-Chinese, negative-English) for each query image and correct them if necessary. If the experts cannot reach consensus on whether a query and reference image match, we discard the query image and replace it with a suitable one from the remaining test data pool, then select new positive and negative reference images for it, followed by model-assisted generation and manual correction of the explanations. This ensures that all test samples are valid and unambiguous, providing a reliable basis for evaluation.
> > > > > >
> > > > > > Among the 1,008 pairs (2,016 explanations), in Round 1, 596 explanations are directly modified and 68 pairs are replaced (34 query images, each with one positive and one negative pair); in Round 2, 48 explanations are directly modified and 8 pairs are replaced (4 query images, each with one positive and one negative pair). In total, 644 direct modifications are made to the explanations, and 76 pairs are replaced (38 query images, each with one positive and one negative pair) across both rounds. The decreasing number of modifications from Round 1 to Round 2 indicates progressive quality improvement and convergence toward high-quality pairs and explanations.
> > > > > >
> > > > > > >***Questions 1: What is the rationale for aligning all modalities exclusively with satellite imagery? Have the authors explored or analyzed alternative anchor modalities, or can they provide evidence (empirical or conceptual) supporting this design choice?***
> > > > > >
> > > > > > Please refer to the response to Weaknesses 1.
> > > > > >
> > > > > > >***Questions 2: Beyond combining existing components such as InfoNCE loss and OpenCLIP-style DDP, what specific algorithmic or architectural innovation does GLEAM introduce? How does it advance the state of the art beyond prior CVGL frameworks?***
> > > > > >
> > > > > > Please refer to the response to Weaknesses 2.
> > > > > >
> > > > > > >***Questions 3: Since GPT-generated explanations serve as the primary supervision and evaluation reference, how do the authors ensure that the model learns genuine interpretive reasoning rather than merely imitating GPT-style phrasing? Were any human-centric or factual quality metrics used?***
> > > > > >
> > > > > > Please refer to the response to Weakness 3, where we discuss two key points: (1) how we ensure the model learns genuine interpretive reasoning rather than merely imitating GPT-style phrasing, and (2) the human-centric and factual quality metrics we employ for evaluation.
> > > > > >
> > > > > > >***Questions 4: Why was VIGOR chosen as the initial dataset in the two-phase training process? Have the authors verified that starting with VIGOR (instead of other datasets) leads to measurable benefits in generalization or convergence?***
> > > > > >
> > > > > > Please refer to the response to Weaknesses 4.1.
> > > > > >
> > > > > > >***Questions 5: Could the authors report inter-annotator agreement scores or qualitative statistics (e.g., number of samples revised or rejected) to substantiate the reliability of the human-refined explanation dataset?***
> > > > > >
> > > > > > Please refer to the response to Weaknesses 4.2.

---

> > > > > > > ### Author Response · Authors · 2025-11-18
> > > > > > >
> > > > > > > Dear Reviewer kFbq,
> > > > > > >
> > > > > > > I hope this message finds you well. As the rebuttal discussion period progresses, I wanted to gently follow up on the responses we submitted for our ICLR submission. We have carefully addressed all the concerns raised in your reviews and would greatly appreciate any feedback or further questions you may have. We remain available for any additional clarification that might be helpful. Thank you very much for your time and consideration.
> > > > > > >
> > > > > > > Best regards,
> > > > > > >
> > > > > > > Authors

---

> > > > > > > > ### Author Response · Authors · 2025-11-24
> > > > > > > >
> > > > > > > > Dear Reviewer kFbq,
> > > > > > > >
> > > > > > > > We hope this message finds you well. We wanted to follow up on the rebuttal we submitted over a week ago for our ICLR submission.
> > > > > > > >
> > > > > > > > We know the review period keeps you busy, and we really appreciate the time you've put into reviewing our paper. We've worked hard to address the concerns from your review. Whenever you have a moment, we'd love to hear what you think about our responses.
> > > > > > > >
> > > > > > > > If you need us to clarify anything else, just let us know. Thanks so much for your time.
> > > > > > > >
> > > > > > > > Best regards,
> > > > > > > >
> > > > > > > > Authors

---

> ### Comment · Area_Chair_vgAn · 2025-11-25
> **Gentle reminder of discussion period**
>
> Dear Reviewer, thank you for reviewing for ICLR. Since the discussion deadline is coming soon, could you please take a look at the author's rebuttal, respond to their comments, and update your rating as well? Thanks!

---

> > ### Author Response · Authors · 2025-11-26
> >
> > Dear Reviewer kFbq,
> >
> > We hope this message finds you well. We are following up on the rebuttal submitted for our ICLR submission.
> >
> > We sincerely appreciate your time and effort in reviewing our work. We would greatly value your feedback at your earliest convenience, as a timely response would help us address any remaining issues effectively before the final deadline.
> >
> > If any points require further clarification, we are happy to provide additional information promptly.
> >
> > Thank you very much for your consideration.
> >
> > Best regards,
> >
> > Authors

---

> > > ### Author Response · Authors · 2025-11-28
> > >
> > > Dear Reviewer kFbq,
> > >
> > > We hope this message finds you well. We are following up on the rebuttal submitted for our ICLR submission.
> > >
> > > We sincerely appreciate your time and effort in reviewing our work. We would greatly value your feedback at your earliest convenience, as a timely response would help us address any remaining issues effectively before the final deadline.
> > >
> > > If any points require further clarification, we are happy to provide additional information promptly.
> > >
> > > Thank you very much for your consideration.
> > >
> > > Best regards,
> > >
> > > Authors

---

### Official Review · Reviewer_gY16 · 2025-10-31

**Soundness:** 3
**Presentation:** 3
**Contribution:** 3
**Rating:** 8
**Confidence:** 4

**Summary:**

This paper introduces GLEAM, a two-component framework that addresses the challenges of multi-view/modal integration and interpretability in CVGL. The first component aligns UAV imagery, street maps, panoramic views, and ground photographs with satellite images through contrastive learning with an efficient two-phase training strategy. The second component establishes a bilingual benchmark combining image matching with explainable reasoning, where a fine-tuned vision-language model outperforms existing systems and generates human-aligned explanations. Overall, GLEAM integrates accurate cross-view matching with interpretable reasoning.

**Strengths:**

## Strengths：

1. **Well-motivated problem.** The paper addresses two important and existing challenges in the CVGL field: limited view/modality coverage and lack of interpretability, emphasizing their practical significance.

2. **Sound technical design.** The proposed two-phase training strategy effectively balances data complexity across datasets, and distributed optimization improves training efficiency.

3. **High-quality benchmark.** By combining large-scale LLM-generated explanations with selected expert refinement, the benchmark ensures both data quality and bilingual generality.

4. **Comprehensive experimental analysis.** Experiments cover multiple evaluation aspects and provide valuable insights into model behavior under different supervision settings.

**Weaknesses:**

## Weaknesses：

I believe the paper's methodology and experiments are thorough and well-detailed. However, the following points could be addressed to further strengthen the work:

1. **Paper structure.** The notion of an "integrated pipeline" for GLEAM-C and GLEAM-X is highlighted as a key contribution, but its operational workflow is not described in a dedicated section. Providing a clear explanation of this operational workflow would help substantiate its significance.

2. **Experimental analysis.** While the paper identifies a strong bias toward negative samples in the original MLLM, the root cause is not investigated. Understanding the origin of the performance discrepancy between positive and negative pairs would clarify this behavior.

3. **Practical usage.** Although the paper discusses the unified model's deployment advantages, it lacks analysis of practical performance metrics. Reporting the model's memory footprint and power consumption would be valuable for real-world deployment.

**Questions:**

## Questions：

1. Could the authors consider adding a dedicated section for the GLEAM-C/X pipeline in the main text to clarify its operational significance?

2. What is the root cause of Qwen2.5-VL-3B's bias, and has its origin been explored?

3. Could the authors provide information on the model's memory footprint and power consumption under real deployment scenarios?

4. Algorithm 1, detailing the DDP optimization, is important for understanding the methodology. Would the authors consider moving it from the Appendix to the main text?

---

> ### Author Response · Authors · 2025-11-16
>
> We are grateful for your positive assessment and insightful suggestions.
>
> >***Weaknesses 1: Paper structure. The notion of an "integrated pipeline" for GLEAM-C and GLEAM-X is highlighted as a key contribution, but its operational workflow is not described in a dedicated section. Providing a clear explanation of this operational workflow would help substantiate its significance.***
>
> Thank you for this suggestion. We add a dedicated subsection in the revised manuscript to clearly illustrate how GLEAM-C and GLEAM-X work together in the integrated pipeline, including the candidate passing mechanism and decision-making process between the two stages.
>
> >***Weaknesses 2: Experimental analysis. While the paper identifies a strong bias toward negative samples in the original MLLM, the root cause is not investigated. Understanding the origin of the performance discrepancy between positive and negative pairs would clarify this behavior.***
>
> Thank you for this insightful observation. The root cause of the negative bias stems from the inherent asymmetry in the verification task itself. Identifying mismatched pairs (negatives) requires finding just one distinguishing feature—a different building style, road pattern, or terrain type is sufficient to conclude the images don't match. In contrast, confirming matched pairs (positives) demands comprehensive evidence across multiple aspects: buildings, roads, vegetation, and overall layout must all align. This asymmetry makes negative judgments naturally easier for MLLMs, leading to the observed bias.
>
> >***Weaknesses 3: Practical usage. Although the paper discusses the unified model's deployment advantages, it lacks analysis of practical performance metrics. Reporting the model's memory footprint and power consumption would be valuable for real-world deployment.***
>
> Thank you for this practical suggestion. We conduct deployment tests on an NVIDIA Jetson AGX Xavier (32GB RAM, JetPack 5.1) to evaluate real-world performance metrics:
>
> | **Module** | **Model** | **Configuration** | **Peak Power (W)** | **Peak Memory (GB)** | **Inference Time (s)** |
> |------------|-----------|-------------------|--------------------|-----------------------|------------------------|
> | GLEAM-C    | ConvNeXt-B-384 | 1 query + 10 references | 30.4 | 5.4 | 4.3 |
> | GLEAM-X    | Qwen2.5-VL-3B-Instruct | 1 query + 1 reference | 35.7 | 12.1 | 28.5 |
>
> These measurements demonstrate that our unified pipeline can run efficiently on edge devices with moderate computational resources. All measurements are obtained under full load conditions without quantization or optimization, indicating potential for further performance improvements in production deployments.
>
> >***Questions 1: Could the authors consider adding a dedicated section for the GLEAM-C/X pipeline in the main text to clarify its operational significance?***
>
> Thank you for this suggestion. As mentioned in our response to Weakness 1, we have added a dedicated section in the revised manuscript.
>
> >***Questions 2: What is the root cause of Qwen2.5-VL-3B's bias, and has its origin been explored?***
>
> Thank you for this question. As mentioned in our response to Weakness 2, we have investigated the cause and identify that the negative bias stems from the inherent asymmetry in the verification task itself.
>
> >***Questions 3: Could the authors provide information on the model's memory footprint and power consumption under real deployment scenarios?***
>
> Thank you for this question. As mentioned in our response to Weakness 3, we conduct deployment tests on an NVIDIA Jetson AGX Xavier to evaluate real-world performance metrics including peak power consumption and memory footprint.
>
> >***Questions 4: Algorithm 1, detailing the DDP optimization, is important for understanding the methodology. Would the authors consider moving it from the Appendix to the main text?***
>
> Thank you for this suggestion. We appreciate the importance of Algorithm 1 for understanding our methodology. However, due to space constraints in the main text, we keep it in the Appendix where it remains easily accessible as the Appendix is directly connected to the main manuscript. We believe the current placement maintains readability while ensuring all technical details are available to interested readers.

---

> > ### Author Response · Authors · 2025-11-24
> >
> > Dear Reviewer gY16,
> >
> > We hope this message finds you well. We wanted to follow up on the rebuttal we submitted over a week ago for our ICLR submission.
> >
> > We know the review period keeps you busy, and we really appreciate the time you've put into reviewing our paper. We've worked hard to address the concerns from your review. Whenever you have a moment, we'd love to hear what you think about our responses.
> >
> > If you need us to clarify anything else, just let us know. Thanks so much for your time.
> >
> > Best regards,
> >
> > Authors

---

> > > ### Comment · Reviewer_gY16 · 2025-11-24
> > >
> > > The responses and revisions of the authors have addressed my concerns. The deployment tests on Jetson AGX Xavier are particularly valuable and make the paper more comprehensive. I believe this work will provide valuable insights to the remote sensing community. Therefore, I will maintain my score.

---

> > > > ### Author Response · Authors · 2025-11-24
> > > >
> > > > Dear Reviewer gY16,
> > > >
> > > > Thank you so much for taking the time to review our responses and for your positive feedback. We're really glad that the deployment tests on Jetson AGX Xavier helped address your concerns. We truly appreciate your thoughtful engagement throughout the review process, and your feedback has been incredibly valuable in improving our work.
> > > >
> > > > Best regards,
> > > >
> > > > Authors

---

### Official Review · Reviewer_bDAZ · 2025-11-02

**Soundness:** 2
**Presentation:** 2
**Contribution:** 2
**Rating:** 4
**Confidence:** 4

**Summary:**

This paper introduces a unified framework for cross-view geo-localization (CVGL) that integrates both matching and explainability. The proposed system comprises two complementary components: GLEAM-C, a foundational CVGL model aligning multiple modalities with satellite imagery using a two-phase training strategy and distributed data parallel optimization; and GLEAM-X, an explainable benchmark that employs multimodal large language models to produce interpretable reasoning for image correspondence. By constructing a bilingual dataset with human-refined annotations and leveraging models like GPT-4o and Doubao-1.5, GLEAM-X enables systematic evaluation of explainable matching.

**Strengths:**

1. The paper makes a novel contribution by combining multi-modal alignment with explainable reasoning in CVGL. The integration of MLLM-based explanation into visual matching tasks (GLEAM-X) represents a creative and timely advancement, addressing interpretability, a known limitation in existing CVGL models.
2. Experimental results across multiple datasets (VIGOR, University-1652, SetVL-480K, MAP) are comprehensive, and the reported efficiency gains are well-quantified.
3. The paper is well-structured. Figures and tables effectively illustrate architecture and results.

**Weaknesses:**

1. Limited novelty in the core CVGL model: While GLEAM-C unifies modalities effectively, its architecture largely builds on prior contrastive learning paradigms without introducing new architectural elements beyond training optimizations.
2. GLEAM-X depends heavily on GPT-4o and Doubao-generated labels, which may introduce linguistic bias or superficial reasoning patterns. The evaluation of “explanation quality” via semantic similarity does not fully capture human interpretability or factual grounding.
3. Main figures not show enough information of the work (dataset catories, detailed data distribution), and are not informative enough.

**Questions:**

1. Human Evaluation: Have human judges been used to assess the fidelity and usefulness of generated explanations, beyond Sentence-BERT similarity? How consistent are these with human rationales?
2. Generalization: How well does GLEAM-C perform when trained on one modality and tested on unseen modalities? A cross-domain generalization study could highlight robustness.
3. Bias Mitigation: Since GPT-based models generated training explanations, what measures were taken to ensure linguistic fairness and minimize cultural/geographical bias in the bilingual dataset?

---

> ### Author Response · Authors · 2025-11-16
>
> We thank you for your detailed review and acknowledge your concerns.
>
> >***Weaknesses 1: Limited novelty in the core CVGL model: While GLEAM-C unifies modalities effectively, its architecture largely builds on prior contrastive learning paradigms without introducing new architectural elements beyond training optimizations.***
>
> We respectfully disagree with the assessment that GLEAM-C offers "limited novelty" or merely "training optimizations." Our contributions span multiple dimensions beyond architectural design.
>
> First, GLEAM-C is the first unified CVGL model that handles four heterogeneous views/modalities (UAV, maps, panoramas, ground images) through a single architecture by aligning them exclusively with satellite imagery. Unlike prior work that develops view/modality-specific models, our unified design offers deployment efficiency with a single inference interface for different views and modalities, reducing memory footprint and initialization latency for multi-platform deployment while maintaining strong performance across views and modalities.
>
> Second, we propose a two-phase training strategy as a methodological innovation to address the challenge of training on multiple datasets with varying sizes and difficulty levels. Phase 1 trains on VIGOR for 40 epochs to establish fundamental CVGL capabilities, while Phase 2 trains on concatenated multi-view/modal datasets for 40 epochs to enable cross-view/modal localization. As shown in Tab. 2, compared to direct mixed-data training (from-scratch merge), our two-phase strategy achieves better performance. Compared to training separate models for each view/modality (single-dataset training), our unified model with two-phase training also attains comparable or superior results across all datasets.
>
> Third, we reconstruct the training paradigm from DP to DDP with global feature gathering, which is critical for enabling large-scale multi-modal CVGL training. By implementing feature gathering, each GPU collects features from all other GPUs before loss computation, preserving gradient flow while providing sufficient negative samples for contrastive learning. As shown in Tab. 3, this achieves 5x training speedup (521s to 92s per epoch) with no accuracy degradation, making multi-modal training practically feasible at scale.
>
> Finally, our comprehensive evaluation across multiple backbone architectures, datasets, views and modalities demonstrates that a unified model can match modality-specific models' performance, which is non-trivial given data heterogeneity. These contributions collectively represent a significant methodological and practical advancement in CVGL, not merely training optimizations.
>
> It's worth clarifying that innovation in machine learning extends beyond introducing novel "architectural elements". Prior contrastive learning paradigms are tools, just like CNNs or transformers. The research value lies in how we design frameworks, devise training strategies, and integrate components to address new challenges. Our work demonstrates that thoughtful system design and methodological innovation in unifying heterogeneous views and modalities, developing effective multi-dataset training strategies, and enabling scalable distributed training constitute substantial research contributions that advance the field beyond incremental architectural modifications.

---

> > ### Author Response · Authors · 2025-11-16
> >
> > >***Weaknesses 2.1: GLEAM-X depends heavily on GPT-4o and Doubao-generated labels, which may introduce linguistic bias or superficial reasoning patterns.***
> >
> > Thank you for this important concern. We acknowledge that knowledge distillation inherently carries the risk of propagating teacher limitations to student models. Our approach addresses these issues as follows:
> >
> > For linguistic bias: Ground-truth guidance during data curation directly mitigates this. When generating explanations, we provide GT matching labels to teacher models, steering them toward correct reasoning directions. Tab. 4-5 show that teacher models (GPT-4o and Doubao-1.5) achieve 77-84% average accuracy and 0.74-0.83 average similarity when evaluated without GT labels during inference. In contrast, student models (Qwen2.5-VL-3B) trained on GT-guided explanations achieve 87-91% accuracy and 0.79-0.84 similarity, demonstrating that GT supervision effectively controls bias and enables the student to even surpass teacher performance.
> >
> > For superficial reasoning patterns: We acknowledge that this depends on the teacher model's inherent capabilities, which determine the quality ceiling of generated explanations. In this work, we have not performed additional data refinement (e.g., human filtering or iterative improvement). However, student models maintain 0.79-0.84 average similarity with ground-truth explanations (Tab. 5) and achieve strong matching performance (Tab. 4), demonstrating that the current explanation quality is sufficient for effective knowledge transfer. As this is the first work to introduce explanation supervision for cross-view geo-localization, improving data quality through advanced curation strategies remains a promising direction for future work.
> >
> > >***Weaknesses 2.2: The evaluation of “explanation quality” via semantic similarity does not fully capture human interpretability or factual grounding.***
> >
> > We agree with you that Sentence-BERT alone may miss critical reasoning errors. We add two additional evaluation methods (EN) to address this concern.
> >
> > First, we conduct an LLM-as-a-judge evaluation using Gemini 2.5 Flash with a detailed 0-5 scoring rubric focusing on identifying geographical features and spatial reasoning accuracy. We choose Gemini 2.5 Flash because it is independent from the training data sources and less likely to show preference bias. Importantly, we provide Gemini with the query image, reference image, reference explanation, and model-generated explanation, enabling it to assess factual correctness of the reasoning rather than just linguistic quality. Second, we perform a human evaluation using the same rubric with 4 human experts in Remote Sensing. To improve consistency in human evaluation, each expert evaluates one complete dataset among MAP, SetVL-480K, University-1652, and VIGOR. The detailed scoring rubric is provided in the Appendix.
> >
> > The results below show consistent patterns across all three metrics. In the LLM-as-a-judge evaluation, the explanation-supervised models achieve average scores of 3.68 and 3.65, even surpassing their teacher models GPT-4o (3.27) and Doubao-1.5 (3.25), while substantially outperforming the baseline Qwen2.5-VL-3B-Instruct at 1.77. The human evaluation confirms this trend with scores of 3.09 and 3.11 versus 2.73 and 2.79 for the teachers and 1.64 for the baseline. The label-only model scores near zero (0.35 and 0.63) as it cannot generate explanations. These results align well with the Sentence-BERT scores in Tab. 5 of the paper where the explanation-supervised models show similar improvements. The consistency across semantic similarity, LLM judgment, and human assessment demonstrates that the explanation-supervised models produce high-quality reasoning.
> >
> >
> > | Gemini   2.5 Flash |  |  |  |  |  |  |  |
> > |---|:---:|:---:|:---:|:---:|:---:|:---:|:---:|
> > | Model | 0 | 1 | 2 | 3 | 4 | 5 | Avg Score |
> > | GPT-4o | 17.5% | 0.2% | 1.6% | 7.5% | 65.2% | 8.0% | 3.27  |
> > | Doubao-1.5 | 19.5% | 0.0% | 1.0% | 4.1% | 66.0% | 9.4% | 3.25  |
> > | Qwen2.5-VL-3B-Instruct | 42.2% | 2.3% | 17.0% | 14.3% | 23.2% | 1.1% | 1.77  |
> > | Qwen2.5-VL-3B+label only | 90.7% | 0.2% | 0.5% | 0.8% | 7.4% | 0.4% | 0.35  |
> > | Qwen2.5-VL-3B+GPT-4o | 8.7% | 0.2% | 3.0% | 6.6% | 65.2% | 16.3% | 3.68  |
> > | Qwen2.5-VL-3B+Doubao-1.5 | 8.6% | 0.1% | 1.9% | 10.9% | 63.7% | 14.8% | 3.65  |
> >
> >
> > | Human Test |  |  |  |  |  |  |  |
> > |---|:---:|:---:|:---:|:---:|:---:|:---:|:---:|
> > | Model | 0 | 1 | 2 | 3 | 4 | 5 | Avg Score |
> > | GPT-4o | 11.9% | 9.4% | 20.8% | 15.4% | 36.2% | 6.3% | 2.73  |
> > | Doubao-1.5 | 13.4% | 10.4% | 15.7% | 12.4% | 40.2% | 7.9% | 2.79  |
> > | Qwen2.5-VL-3B-Instruct | 28.3% | 22.3% | 24.0% | 9.4% | 14.1% | 1.9% | 1.64  |
> > | Qwen2.5-VL-3B+label only | 66.5% | 13.4% | 14.4% | 2.5% | 2.8% | 0.5% | 0.63  |
> > | Qwen2.5-VL-3B+GPT-4o | 6.6% | 8.1% | 19.1% | 13.7% | 41.5% | 11.1% | 3.09  |
> > | Qwen2.5-VL-3B+Doubao-1.5 | 5.8% | 6.9% | 20.6% | 15.4% | 39.5% | 11.8% | 3.11  |

---

> > > ### Author Response · Authors · 2025-11-16
> > >
> > > >***Weaknesses 3: Main figures do not show enough information of the work (dataset categories, detailed data distribution), and are not informative enough.***
> > >
> > > We respectfully disagree with this concern. Our paper provides comprehensive information about dataset composition and distribution:
> > >
> > > (1) Dataset Categories and Distribution (Tab. 1). Table 1 explicitly presents: Four dataset categories (University-1652, VIGOR, SetVL-480K, and MAP); Original pair counts for each dataset (ranging from 10,208 to 240,544); Sampling ratios showing how GLEAM-C and GLEAM-X balance data distribution (e.g., University-1652: 1.00 for GLEAM-C and 0.36 for GLEAM-X, MAP: 4.00 for GLEAM-C and 1.00 for GLEAM-X); Final training pair counts for both GLEAM-C (251,567) and GLEAM-X (202,832). This clearly demonstrates our data balancing strategy across datasets of varying scales.
> > >
> > > (2) Detailed Performance Breakdown (Tab. 2-6). Our analysis tables provide granular breakdowns across all four datasets, both languages (English and Chinese), and positive vs. negative pairs (Pos Acc, Neg Acc, Pos-Neg gap). For example, Tab. 6 reveals critical insights like the negative bias in original models (Pos-Neg gap: -67.06 for EN, -98.61 for ZH) and its mitigation through training (+2.18).
> > >
> > > (3) Sampling Strategy. The "Sampling Ratio" column in Tab. 1 directly reflects our balanced random sampling approach to ensure fair representation across datasets with different scales, preventing dominance by larger datasets like SetVL-480K.
> > >
> > > We believe these tables collectively provide sufficient detail on dataset composition, distribution, and performance characteristics. If you seek specific additional visualizations (e.g., pie charts of dataset proportions), we are happy to add them in the revision.
> > >
> > > >***Questions 1: Human Evaluation: Have human judges been used to assess the fidelity and usefulness of generated explanations, beyond Sentence-BERT similarity? How consistent are these with human rationales?***
> > >
> > > As mentioned in the response to Weaknesses 2.2, we additionally conduct LLM-as-a-judge evaluation and human evaluation.

---

> > > > ### Author Response · Authors · 2025-11-16
> > > >
> > > > >***Questions 2: Generalization: How well does GLEAM-C perform when trained on one modality and tested on unseen modalities? A cross-domain generalization study could highlight robustness.***
> > > >
> > > > Thank you for this valuable suggestion. We have conducted cross-domain generalization experiments to evaluate the model's robustness across different modalities.
> > > >
> > > > **Experimental Setup:** We first pretrain the model (ConvNeXt-B-384) on VIGOR only and test it on all four datasets without any fine-tuning. We then continue training on VIGOR+MAP to examine whether incremental learning improves generalization.
> > > >
> > > > #### **Scenario 1: Zero-shot Transfer (VIGOR pretrain only)**
> > > > | Pretrain Dataset | Test Dataset    | R@1   | R@5   | R@10  | Top-1 | AP/HR |
> > > > | ---------------- | --------------- | ----- | ----- | ----- | ----- | ----- |
> > > > | VIGOR            | VIGOR           | 76.60 | 95.07 | 96.83 | 99.64 | 88.29 |
> > > > | VIGOR            | University-1652 | 19.93 | 37.37 | 46.59 | 47.89 | 24.43 |
> > > > | VIGOR            | MAP             | 0.74  | 2.04  | 2.74  | 11.32 | 0.74  |
> > > > | VIGOR            | SetVL           | 0.18  | 0.65  | 1.12  | 4.68  | 0.18  |
> > > >
> > > > #### **Scenario 2: Incremental Training (VIGOR pretrain → VIGOR + MAP fine-tune)**
> > > > | Training Strategy | Test Dataset    | R@1   | R@5   | R@10  | Top-1 | AP/HR |
> > > > | ----------------- | --------------- | ----- | ----- | ----- | ----- | ----- |
> > > > | VIGOR → VIGOR+MAP | VIGOR           | 77.29 | 95.25 | 96.90 | 99.66 | 88.85 |
> > > > | VIGOR → VIGOR+MAP | University-1652 | 10.02 | 20.80 | 27.67 | 28.78 | 13.12 |
> > > > | VIGOR → VIGOR+MAP | MAP             | 93.85 | 98.35 | 98.79 | 99.88 | 93.85 |
> > > > | VIGOR → VIGOR+MAP | SetVL           | 0.04  | 0.24  | 0.44  | 1.92  | 0.04  |
> > > >
> > > > **Results and Analysis:**
> > > >
> > > > The zero-shot transfer results reveal significant challenges in cross-modality generalization. When pretrained solely on VIGOR, the model achieves 76.60% R@1 on VIGOR but drops dramatically to 0.74% on MAP and 0.18% on SetVL. This substantial performance degradation stems from fundamental differences in viewpoint characteristics. These results indicate that features learned from one modality do not transfer well to drastically different view types without explicit training.
> > > >
> > > > Incremental training on VIGOR+MAP shows that performance improves only for datasets included in training. MAP performance increases significantly from 0.74% to 93.85% R@1, while VIGOR maintains comparable performance (77.29% vs. 76.60%). However, unseen modalities experience negative transfer: University-1652 drops from 19.93% to 10.02%, and SetVL decreases from 0.18% to 0.04%. This suggests that simply adding one modality does not enable generalization to other unseen view types.
> > > >
> > > > In contrast, our unified framework with multi-dataset training (Tab. 2 of paper) demonstrates that when all modalities are jointly trained, the model achieves performance comparable to or better than single-dataset training. For instance, using PE-Core-L14-336 backbone, the two-phase training strategy achieves 75.96% R@1 on VIGOR (vs. 75.53% single-dataset), 93.97% on MAP (vs. 92.79%), and 23.25% on SetVL (vs. 21.34%). This validates that our architecture can effectively leverage diverse views and modalities when they are available during training, avoiding the negative transfer observed in incremental learning scenarios.
> > > >
> > > > These findings highlight that while zero-shot cross-modality transfer remains challenging due to inherent viewpoint gaps, our unified approach successfully handles multiple modalities simultaneously when trained jointly, which is one of the primary contributions of our work.
> > > >
> > > > >***Questions 3: Bias Mitigation: Since GPT-based models generated training explanations, what measures were taken to ensure linguistic fairness and minimize cultural/geographical bias in the bilingual dataset?***
> > > >
> > > > Thank you for this important question. To address potential biases, we generate the Chinese and English explanations simultaneously in a single prompt, explicitly asking the model to provide corresponding descriptions in both languages. This approach ensures semantic alignment from the start.
> > > >
> > > > More importantly, the nature of geographic localization naturally limits cultural bias. The explanations focus on objective, observable features like building density, road patterns, terrain types, and infrastructure layouts—things like "high-rise buildings," "river intersections," or "highway interchanges." These are factual geographic descriptions rather than subjective cultural interpretations.
> > > >
> > > > We acknowledge that GPT models may have inherent biases from their training data. However, because our task centers on matching physical geographic features, the impact of such biases is substantially reduced. The descriptions are grounded in what's visually present in the images rather than cultural context or value judgments.

---

> > > > > ### Author Response · Authors · 2025-11-18
> > > > >
> > > > > Dear Reviewer bDAZ,
> > > > >
> > > > > I hope this message finds you well. As the rebuttal discussion period progresses, I wanted to gently follow up on the responses we submitted for our ICLR submission. We have carefully addressed all the concerns raised in your reviews and would greatly appreciate any feedback or further questions you may have. We remain available for any additional clarification that might be helpful. Thank you very much for your time and consideration.
> > > > >
> > > > > Best regards,
> > > > >
> > > > > Authors

---

> > > > > > ### Author Response · Authors · 2025-11-24
> > > > > >
> > > > > > Dear Reviewer bDAZ,
> > > > > >
> > > > > > We hope this message finds you well. We wanted to follow up on the rebuttal we submitted over a week ago for our ICLR submission.
> > > > > >
> > > > > > We know the review period keeps you busy, and we really appreciate the time you've put into reviewing our paper. We've worked hard to address the concerns from your review. Whenever you have a moment, we'd love to hear what you think about our responses.
> > > > > >
> > > > > > If you need us to clarify anything else, just let us know. Thanks so much for your time.
> > > > > >
> > > > > > Best regards,
> > > > > >
> > > > > > Authors

---

> ### Comment · Area_Chair_vgAn · 2025-11-25
> **Gentle reminder on discussion period**
>
> Dear Reviewer, thank you for reviewing for ICLR. Since the discussion deadline is coming soon, could you please take a look at the author's rebuttal, respond to their comments, and update your rating as well? Thanks!

---

> > ### Author Response · Authors · 2025-11-26
> >
> > Dear Reviewer bDAZ,
> >
> > We hope this message finds you well. We are following up on the rebuttal submitted for our ICLR submission.
> >
> > We sincerely appreciate your time and effort in reviewing our work. We would greatly value your feedback at your earliest convenience, as a timely response would help us address any remaining issues effectively before the final deadline.
> >
> > If any points require further clarification, we are happy to provide additional information promptly.
> >
> > Thank you very much for your consideration.
> >
> > Best regards,
> >
> > Authors

---

> > > ### Author Response · Authors · 2025-11-28
> > >
> > > Dear Reviewer bDAZ,
> > >
> > > We hope this message finds you well. We are following up on the rebuttal submitted for our ICLR submission.
> > >
> > > We sincerely appreciate your time and effort in reviewing our work. We would greatly value your feedback at your earliest convenience, as a timely response would help us address any remaining issues effectively before the final deadline.
> > >
> > > If any points require further clarification, we are happy to provide additional information promptly.
> > >
> > > Thank you very much for your consideration.
> > >
> > > Best regards,
> > >
> > > Authors

---

### Official Review · Reviewer_hWJr · 2025-11-06

**Soundness:** 3
**Presentation:** 4
**Contribution:** 3
**Rating:** 6
**Confidence:** 4

**Summary:**

The paper presents GLEAM, a comprehensive framework for CVGL. It introduces GLEAM-C, a unified foundational model that aligns multiple modalities with satellite imagery using contrastive learning. Additionally, it proposes GLEAM-X, a novel benchmark for explainable CVGL, leveraging MLLMs to provide interpretable reasoning for matching predictions.

**Strengths:**

* The integration of diverse input modalities—UAV imagery, street maps, panoramic views, and ground-level photos—into a single, satellite-aligned CVGL framework. The definition is clear and has important potential social impacts in real-world scenarios.
* Comprehensive experimental and implementation considerations.

**Weaknesses:**

* Need to add extra experiments for assessing the quality of the generated explanations. Currently, the paper calculates explanations’ semantic similarity to the annotated references with Sentence-BERT. However, Sentence-BERT may fail to capture the key reasoning step. The score would be high when the logic seems to be true, but false in key steps. Adding a user study or LLM-as-a-judge is important.
* As shown in Table 4, the model trained with label-only supervision (+label only) achieves a high matching accuracy of 92.76%, whereas the model trained with explanation supervision (+GPT-4o) drops to 90.38%. Discussion on this is expected. Does this suggest that incorporating explainability may come at the cost of reduced matching performance? However, the paper does not provide an in-depth analysis of this trade-off.
* Insufficient analysis of failure cases. The paper lacks an in-depth analysis of model failure cases and does not discuss the specific scenarios in which the model underperforms or the underlying reasons for such failures.
* The teacher model (GPT4o and Doubao-1.5) may introduce bias during training data curation. Discussing this issue will enhance the rigor of the data curation workflow.

**Questions:**

see above weakness

Others:
- Write the full name of GNSS in the first paragraph that firstly mention it.

---

> ### Author Response · Authors · 2025-11-16
>
> We sincerely appreciate your constructive feedback and valuable suggestions.
>
> >***Weaknesses 1: Need to add extra experiments for assessing the quality of the generated explanations. Currently, the paper calculates explanations’ semantic similarity to the annotated references with Sentence-BERT. However, Sentence-BERT may fail to capture the key reasoning step. The score would be high when the logic seems to be true, but false in key steps. Adding a user study or LLM-as-a-judge is important.***
>
> We agree with you that Sentence-BERT alone may miss critical reasoning errors. We add two additional evaluation methods (EN) to address this concern.
>
> First, we conduct an LLM-as-a-judge evaluation using Gemini 2.5 Flash with a detailed 0-5 scoring rubric focusing on identifying geographical features and spatial reasoning accuracy. We choose Gemini 2.5 Flash because it is independent from the training data sources and less likely to show preference bias. Importantly, we provide Gemini with the query image, reference image, reference explanation, and model-generated explanation, enabling it to assess factual correctness of the reasoning rather than just linguistic quality. Second, we perform a human evaluation using the same rubric with 4 human experts in Remote Sensing. To improve consistency in human evaluation, each expert evaluates one complete dataset among MAP, SetVL-480K, University-1652, and VIGOR. The detailed scoring rubric is provided in the Appendix.
>
> The results below show consistent patterns across all three metrics. In the LLM-as-a-judge evaluation, the explanation-supervised models achieve average scores of 3.68 and 3.65, even surpassing their teacher models GPT-4o (3.27) and Doubao-1.5 (3.25), while substantially outperforming the baseline Qwen2.5-VL-3B-Instruct at 1.77. The human evaluation confirms this trend with scores of 3.09 and 3.11 versus 2.73 and 2.79 for the teachers and 1.64 for the baseline. The label-only model scores near zero (0.35 and 0.63) as it cannot generate explanations. These results align well with the Sentence-BERT scores in Tab. 5 of the paper where the explanation-supervised models show similar improvements. The consistency across semantic similarity, LLM judgment, and human assessment demonstrates that the explanation-supervised models produce high-quality reasoning.
>
>
> | Gemini   2.5 Flash |  |  |  |  |  |  |  |
> |---|:---:|:---:|:---:|:---:|:---:|:---:|:---:|
> | Model | 0 | 1 | 2 | 3 | 4 | 5 | Avg Score |
> | GPT-4o | 17.5% | 0.2% | 1.6% | 7.5% | 65.2% | 8.0% | 3.27  |
> | Doubao-1.5 | 19.5% | 0.0% | 1.0% | 4.1% | 66.0% | 9.4% | 3.25  |
> | Qwen2.5-VL-3B-Instruct | 42.2% | 2.3% | 17.0% | 14.3% | 23.2% | 1.1% | 1.77  |
> | Qwen2.5-VL-3B+label only | 90.7% | 0.2% | 0.5% | 0.8% | 7.4% | 0.4% | 0.35  |
> | Qwen2.5-VL-3B+GPT-4o | 8.7% | 0.2% | 3.0% | 6.6% | 65.2% | 16.3% | 3.68  |
> | Qwen2.5-VL-3B+Doubao-1.5 | 8.6% | 0.1% | 1.9% | 10.9% | 63.7% | 14.8% | 3.65  |
>
>
> | Human Test |  |  |  |  |  |  |  |
> |---|:---:|:---:|:---:|:---:|:---:|:---:|:---:|
> | Model | 0 | 1 | 2 | 3 | 4 | 5 | Avg Score |
> | GPT-4o | 11.9% | 9.4% | 20.8% | 15.4% | 36.2% | 6.3% | 2.73  |
> | Doubao-1.5 | 13.4% | 10.4% | 15.7% | 12.4% | 40.2% | 7.9% | 2.79  |
> | Qwen2.5-VL-3B-Instruct | 28.3% | 22.3% | 24.0% | 9.4% | 14.1% | 1.9% | 1.64  |
> | Qwen2.5-VL-3B+label only | 66.5% | 13.4% | 14.4% | 2.5% | 2.8% | 0.5% | 0.63  |
> | Qwen2.5-VL-3B+GPT-4o | 6.6% | 8.1% | 19.1% | 13.7% | 41.5% | 11.1% | 3.09  |
> | Qwen2.5-VL-3B+Doubao-1.5 | 5.8% | 6.9% | 20.6% | 15.4% | 39.5% | 11.8% | 3.11  |
>
> >***Weaknesses 2: As shown in Table 4, the model trained with label-only supervision (+label only) achieves a high matching accuracy of 92.76%, whereas the model trained with explanation supervision (+GPT-4o) drops to 90.38%. Discussion on this is expected. Does this suggest that incorporating explainability may come at the cost of reduced matching performance? However, the paper does not provide an in-depth analysis of this trade-off.***
>
> The 2.38% accuracy difference is actually expected and reasonable because the explanation-supervised model is solving a much harder problem. The label-only model just does binary classification while the explanation-supervised model must generate full reasoning in natural language while making predictions. As you correctly noted, the longer ground truth makes the model harder to train, but this is necessary to gain explainability. Looking at Tab. 4, the explanation-supervised model at 90.38% still substantially outperforms the original Qwen2.5-VL-3B-Instruct baseline, and more importantly, it can explain its decisions while the label-only model completely cannot do this. We view this as a design choice rather than a limitation where GLEAM-X prioritizes interpretable geo-localization over maximizing matching accuracy alone.

---

> > ### Author Response · Authors · 2025-11-16
> >
> > >***Weaknesses 3: Insufficient analysis of failure cases. The paper lacks an in-depth analysis of model failure cases and does not discuss the specific scenarios in which the model underperforms or the underlying reasons for such failures.***
> >
> > Thank you for this valuable suggestion. Based on our experimental results, we identify following failure patterns:
> >
> > #### 1. For GLEAM-C (Tab. 2)
> >
> > (1) Model Capacity and Training Stability. Smaller backbones (ConvNeXt-B-384) suffer from overtraining in mixed-data settings. Since University-1652 requires only 1 epoch but we train for 40 epochs to accommodate other datasets, ConvNeXt-B-384 shows significant performance drops on University-1652 (87.03 vs. 91.55 R@1 single-dataset), while larger models (PE-Core-L14-336) remain more stable (93.19 vs. 94.38 R@1).
> >
> > (2) GPS Removal for Unified Training. To enable unified training across diverse datasets where GPS information is not available (e.g., University-1652), we remove GPS priors while retaining DSS sampling. As shown in Tab. 2, this design choice leads to performance gaps on some datasets (e.g., VIGOR: 75.66 vs. 77.86 R@1 with GPS), but ensures our framework can handle scenarios where GPS metadata is absent.
> >
> > #### 2. For GLEAM-X
> >
> > (1) Negative Bias of Original Models. As shown in Tab. 6, original MLLMs exhibit significant negative bias across both languages, with substantially higher accuracy on negative pairs than positive pairs (e.g., Qwen2.5-VL-3B-Instruct EN: 23.81% vs. 90.87%, Pos-Neg gap of -67.06; ZH: 1.19% vs. 99.80%, gap of -98.61). This bias stems from task asymmetry: identifying mismatches requires finding just one distinguishing feature, while confirming matches demands comprehensive evidence across multiple aspects. This asymmetry makes negative judgments naturally easier for MLLMs, leading to the observed bias.
> >
> > (2) Data Curation Bias. The quality of generated explanations is constrained by the teacher model's reasoning capabilities, which is a common limitation in distillation-based methods. Without additional data refinement (e.g., human filtering or iterative improvement), certain explanations may exhibit superficial reasoning or fail to capture critical cross-view correspondences. A detailed analysis is provided in the next response.
> >
> > >***Weaknesses 4: The teacher model (GPT4o and Doubao-1.5) may introduce bias during training data curation. Discussing this issue will enhance the rigor of the data curation workflow.***
> >
> > Thank you for this important concern. We acknowledge that teacher models may introduce bias during training data curation. Our approach mitigates this risk through ground-truth guidance.
> >
> > When generating explanations, we provide GT matching labels to teacher models, steering them toward correct reasoning directions. This GT supervision directly controls potential biases from the teacher models. Tab. 4-5 demonstrate the effectiveness of this approach: teacher models (GPT-4o and Doubao-1.5) achieve 77-84% average accuracy and 0.74-0.83 average similarity when evaluated without GT labels during inference. In contrast, student models (Qwen2.5-VL-3B) trained on GT-guided explanations achieve 87-91% accuracy and 0.79-0.84 similarity, demonstrating that GT supervision effectively controls bias and enables the student to even surpass teacher performance.
> >
> > While we have not performed additional data refinement (e.g., human filtering or iterative improvement) in this work, the strong student performance shows that the current explanation quality is sufficient for effective knowledge transfer. As this is the first work to introduce explanation supervision for cross-view geo-localization, improving data quality through advanced curation strategies remains a promising direction for future work.
> >
> > >***Questions 1: Write the full name of GNSS in the first paragraph that firstly mention it.***
> >
> > Thank you for this suggestion. We have added the full name "Global Navigation Satellite System (GNSS)" when it first appears in the introduction.

---

> > > ### Author Response · Authors · 2025-11-24
> > >
> > > Dear Reviewer hWJr,
> > >
> > > We hope this message finds you well. We wanted to follow up on the rebuttal we submitted over a week ago for our ICLR submission.
> > >
> > > We know the review period keeps you busy, and we really appreciate the time you've put into reviewing our paper. We've worked hard to address the concerns from your review. Whenever you have a moment, we'd love to hear what you think about our responses.
> > >
> > > If you need us to clarify anything else, just let us know. Thanks so much for your time.
> > >
> > > Best regards,
> > >
> > > Authors

---

> ### Comment · Area_Chair_vgAn · 2025-11-25
> **kindly reminder on discussion period**
>
> Dear Reviewer, thank you for reviewing for ICLR. Since the discussion deadline is coming soon, could you please take a look at the author's rebuttal, respond to their comments, and update your rating as well? Thanks!

---

> > ### Comment · Reviewer_hWJr · 2025-11-25
> >
> > The authors’ responses and revisions have adequately addressed my concerns. The new experiments evaluating the quality of the generated explanations strengthen the paper’s soundness. I will keep my positive score.

---

> > > ### Author Response · Authors · 2025-11-25
> > >
> > > Dear Reviewer hWJr,
> > >
> > > Thank you very much for your thoughtful feedback and for maintaining your positive assessment of our work. We greatly appreciate your support.
> > >
> > > Best regards,
> > >
> > > Authors

---

### Author Response · Authors · 2025-11-16

We sincerely thank all reviewers for their thorough and constructive reviews.  We have carefully addressed all comments with detailed responses and made corresponding revisions to the manuscript. Below, we summarize the major changes in the paper in sequential order (highlighted in blue) and their correspondence to the reviewers' comments.

1. Line 039:
    - Highlighting "satellite observations" (kFbq)
    - Expanding GNSS acronym to full form (hWJr)
2. Line 269:
    - Adding analysis of linguistic fairness (bDAZ)
3. Line 300:
    - Adding a dedicated section for the GLEAM-C/X pipeline (gY16)
4. Line 317 and 810:
    - Adding real-world deployment performance (gY16)
5. Line 394 and 829:
    - Adding cross-domain generalization experiments of GLEAM-C (bDAZ)
6. Line 394 and 877:
    - Explaining why we chose VIGOR as the initial dataset (kFbq)
7. Line 424:
    - Adding analysis of trade-off between training with explanations and solely matching labels (hWJr)
8. Line 477:
    - Adding analysis of negative bias of original MLLMs (gY16)
9. Line 501 and 922:
    - Adding LLM-as-a-judge and human test (hWJr, bDAZ, kFbq)
10. Line 511:
    - Analyzing potential bias of teacher models (hWJr, bDAZ, kFbq)
11. Line 781:
    - Adding more details of human annotation (kFbq)

---

### Author Response · Authors · 2025-11-27
**Summary for AC, Reviewers bDAZ and kFbq**

Dear AC and Reviewers,

As we approach the final week of the rebuttal period, we provide this brief summary to facilitate your final assessment. We are pleased that **Reviewer hWJr** has confirmed our revisions addressed their concerns , and **Reviewer gY16** maintains a strong positive rating. To address the core concerns raised by **Reviewers bDAZ and kFbq**,  we have submitted our responses and a revised manuscript that substantially strengthens the work in five key areas:

**1) Evaluation Robustness**

We introduced independent verification methods, including an LLM-as-a-judge protocol (using Gemini 2.5) and human expert assessments. These consistently validate that our explanation-supervised models produce high-quality reasoning that surpasses teacher model performance across metrics.

**2) Generalization Capability**

We conducted comprehensive cross-domain experiments. These reveal the inherent challenges of zero-shot transfer across viewpoints and demonstrate the effectiveness of our unified joint training approach in preventing negative transfer.

**3) Data Quality Assurance**

We provided a detailed analysis of our data curation process, emphasizing GT-guided generation, multi-round human revision, and bias mitigation strategies. Our experiments show student models trained with GT-guidance effectively reduce teacher bias.

**4) Design Choice Validation**

We empirically justified the selection of VIGOR as the initial training dataset. Comparative experiments demonstrate that VIGOR offers the optimal balance of difficulty for effective multi-dataset feature initialization.

**5) Contribution Clarification**

We clarified that GLEAM's novelty lies in its system design and methodological advances: it is the first unified framework for multi-view/modal CVGL with explainable reasoning, achieving significant efficiency gains (5x speedup via DDP optimization) and interpretability breakthroughs.

We recognize the significant workload involved in the review process. We have worked diligently to address every concern through additional experiments and manuscript revisions. We would be immensely grateful if Reviewers bDAZ and kFbq could let us know if these revisions have resolved your concerns.

Best regards,

The Authors

---

### Author Response · Authors · 2025-11-29
**Summary of Review-Rebuttal Phase**

Dear Area Chair,

Thank you for handling our submission. We understand this is a busy period, so we provide a concise summary of our rebuttal updates to assist in your final assessment—particularly as Reviewers **@bDAZ** and **@kFbq** have not yet responded to our revisions.

**(I) Core Contribution**

We propose **GLEAM**, a unified pipeline for Cross-View Geo-Localization (CVGL). It introduces **GLEAM-C** to unify four distinct views/modalities (UAV, map, panorama, ground) by aligning them to satellite imagery, and **GLEAM-X**, which leverages Multimodal LLMs to provide interpretable reasoning for *why* images match, moving the field beyond simple retrieval or binary predictions.

**(II) Reviewer Consensus on Strengths**

The reviewers consistently highlight three key strengths:

1.  **Novelty & Motivation:** **@bDAZ** calls the integration of explanations a "creative and timely advancement," and **@gY16** notes it addresses "important and existing challenges" in the field.
2.  **Unified Architecture:** **@kFbq** and **@bDAZ** praise the framework for handling heterogeneous data types, with **@kFbq** noting it "eliminates the need for modality-specific architectures."
3.  **Experimental Rigor:** **@gY16** commends the "comprehensive experimental analysis," and **@hWJr** highlights the "comprehensive experimental and implementation considerations."

**(III) Key Revisions & New Experiments**

We utilize the discussion period to conduct targeted experiments addressing the specific concerns of all reviewers, including those who remain silent.

**1. Validating Explanation Quality (Addressing @hWJr, @kFbq, @bDAZ)**
A shared concern is whether the generated explanations are factually grounded. We add two independent evaluations (Appendix A.7):

*   **LLM-as-a-judge:** Using Gemini 2.5 Flash (independent of our training data), our student model achieves scores surpassing its teacher models.
*   **Human Expert Evaluation:** Remote sensing experts rate our method significantly higher than the baseline.
*   *Outcome:* This empirically proves our model learns genuine reasoning rather than just hallucinating or mimicking style.

**2. Robustness & Generalization (Addressing @bDAZ, @gY16)**

*   **Cross-Domain Transfer:** To address **@bDAZ**’s query on generalization, we add a study (Appendix A.5) showing that our unified training strategy effectively prevents the negative transfer observed in zero-shot or incremental learning settings.
*   **Bias Mitigation:** We address **@gY16**’s concern regarding the "negative bias" (tendency to predict non-matches) in original MLLMs. Our fine-tuning strategy effectively reduces the accuracy gap between positive and negative pairs (Tab. 6).

**3. Justifying Design Choices (Addressing @kFbq, @bDAZ)**

*   **Why VIGOR First?** Responding to **@kFbq**, we provide empirical evidence (Appendix A.6) that starting training with VIGOR offers the optimal difficulty balance compared to other datasets.
*   **Why Satellite Anchor?** We clarify for **@kFbq** that satellite imagery is the only modality possessing both precise geocoordinates and visual surface textures, making it the only viable "anchor" for this task.
*   **Fairness:** We address **@bDAZ**’s concern on linguistic bias by detailing our simultaneous bilingual generation workflow and the objective, feature-based nature of the geographic descriptions.

**4. Real-World Deployment (Addressing @gY16)**

*   Moving beyond theoretical metrics, we deploy the full pipeline on an **NVIDIA Jetson AGX Xavier** edge device (Appendix A.4). The system runs efficiently within memory constraints, demonstrating practical viability.

**(IV) Current Status**

*   **Reviewer @gY16 (Rating: 8):** Maintains a strong acceptance rating, noting the deployment tests make the paper "more comprehensive."
*   **Reviewer @hWJr (Rating: 6):** Confirms that our revisions and new explanation evaluations "adequately address" their concerns.

We believe our additional experiments and revisions effectively resolve the open questions raised by **@bDAZ** and **@kFbq**. We hope this summary aids your final decision.

Best regards,

The Authors

---

### Note · Authors · 2026-01-26

I have read and agree with the venue's withdrawal policy on behalf of myself and my co-authors.

---

### Meta-Review · Area_Chair_hLDe · 2025-12-09

**Summary:**

This paper presents a contribution to CVGL by integrating multi-modal alignment with explainable reasoning. The method incorporates UAV imagery, street maps, panoramas, and ground images into a unified contrastive learning framework aligned with satellite imagery, and adopts a two-phase training strategy. Most reviewers acknowledge the value of unifying diverse modalities and appreciate the interpretability offered by the reasoning component. The main concerns focus on (1) the method’s novelty and (2) the quality of GPT-4o and Doubao-generated labels, which may introduce linguistic bias and superficial reasoning. The second concern has been partially addressed through the additional LLM-as-a-judge evaluations and human studies submitted during the rebuttal. However, concerns regarding method novelty remain largely unresolved, as the two-phase training strategy and DDP appear to be engineering refinements rather than substantive conceptual contributions.

**Reviewer Concerns:**

Two major concerns emerged across the reviews. The first concern involves method novelty, raised by Reviewers bDAZ and kFbq, who argue that the architecture largely follows existing contrastive learning paradigms and does not introduce significant new architectural elements beyond training refinements. Although the authors’ response provides useful clarification and better articulates the design motivation, the overall novelty is insufficient and does not demonstrate substantial methodological innovation. The second concern involves the quality and reliability of GPT-4o and Doubao-generated labels, as well as the limitations of Sentence-BERT for evaluation. This concern is partially resolved, as the authors supplemented the work with LLM-as-a-judge evaluations and human assessments.

**Reviewer Scores:**

Reviewer hWJr gave an initial score of 6 and is likely to keep this score. The reviewer notes that the authors’ responses satisfactorily addressed their concerns but does not indicate an intention to raise the score. Reviewer bDAZ gave an initial score of 4 and is also likely to keep it, as concerns about method novelty persist; the two-phase training strategy and DDP are viewed as engineering-oriented improvements that contribute limited novelty to the community. This reviewer acknowledges that the evaluation concern has been largely addressed through the newly added LLM-as-a-judge and human evaluations. Another reviewer (bDAZ, score 8) gave an initial score of 8 and is expected to maintain this score after discussion. Reviewer kFbq gave an initial score of 4 and is likewise likely to keep it; concerns about method novelty remain, while the evaluation-related concerns are partially addressed by the newly added evidence.

---

### Decision · Program_Chairs · 2026-01-26

Reject